# NeoRL: Efficient Exploration for Nonepisodic RL

**Bhavya Sukhija***, **Lenart Treven, Florian Dörfler, Stelian Coros, Andreas Krause**
ETH Zurich, Switzerland

## Abstract

We study the problem of nonepisodic reinforcement learning (RL) for nonlinear dynamical systems, where the system dynamics are unknown and the RL agent has to learn from a single trajectory, i.e., adapt online and without resets. This setting is ubiquitous in the real world, where resetting is impossible or requires human intervention. We propose ***Nonepisodic Optimistic RL** (NeoRL)*, an approach based on the principle of optimism in the face of uncertainty. NeoRL uses well-calibrated probabilistic models and plans optimistically w.r.t. the epistemic uncertainty about the unknown dynamics. Under continuity and bounded energy assumptions on the system, we provide a first-of-its-kind regret bound of $\mathcal{O}(\beta_T \sqrt{T\Gamma_T})$ for general nonlinear systems with Gaussian process dynamics. We compare NeoRL to other baselines on several deep RL environments and empirically demonstrate that NeoRL achieves the optimal average cost while incurring the least regret.

## 1 Introduction

In recent years, data-driven control approaches, such as reinforcement learning (RL), have demonstrated remarkable achievements. However, most RL algorithms are devised for an episodic setting, where during each episode, the agent interacts in the environment for a predetermined episode length or until a termination condition is met. After the episode, the agent is reset back to an initial state from where the next episode commences. Episodes prevent the system from blowing up, i.e., maintain stability, while also restricting exploration to states that are relevant to the task at hand. Moreover, resets ensure that the agent explores close to the initial states and does not end up at undesirable parts of the state space that exhibit low reward. In simulation, resetting is typically straightforward. However, if we wish to enable agents to learn and adapt by interacting online with the real world, resets are often prohibitive since they typically involve manual intervention. Instead, agents should be able to learn autonomously (Sharma et al., 2021b) i.e., from a single trajectory. This problem is extensively studied in adaptive control (Åström & Wittenmark, 2013), where classical works focus on controller design (Lai & Wei, 1982, 1987; Krstić et al., 1992, 1995; Annaswamy, 2023) and not on the exploration/learning aspect of the problem. Only a few works consider these two aspects jointly (Abbasi-Yadkori & Szepesvári, 2011; Cohen et al., 2019; Dean et al., 2020; Simchowitz & Foster, 2020; Zhao et al., 2024). However, these works study linear systems with quadratic costs, i.e., the LQR setting. While several works in the Deep RL community have also studied this problem, (c.f., Section 5), the theoretical results for this setting are fairly limited. In particular, theoretical results mostly exist for the finite state and action spaces (Kearns & Singh, 2002; Brafman & Tennenholtz, 2002; Jaksch et al., 2010) and the extension to nonlinear systems with continuous spaces is much less understood. In our work, we address this gap and propose a practical RL algorithm that is grounded in theory. In particular, we make the following contributions.

**Contributions**

1. We propose, NeoRL, a novel model-based RL algorithm based on the principle of optimism in the face of uncertainty. NeoRL operates in a nonepisodic setting and picks average cost optimal policies optimistically w.r.t. to the model's epistemic uncertainty.

*Correspondence to `sukhijab@ethz.ch`

38th Conference on Neural Information Processing Systems (NeurIPS 2024).

2. We show that when the dynamics lies in a reproducing kernel Hilbert space (RKHS) of kernel $k$, NeoRL exhibits a regret of $\mathcal{O}(\beta_T \sqrt{T\Gamma_T})$, where the regret, akin to prior work, is measured w.r.t to the optimal average cost under known dynamics, $T$ is the number of environment steps, $\beta_T$ the calibration coefficient (Chowdhury & Gopalan, 2017; Srinivas et al., 2012) and $\Gamma_T$ the maximum information gain of kernel $k$ (Srinivas et al., 2012). Our regret bound is similar to the ones obtained in the episodic setting (Kakade et al., 2020; Curi et al., 2020; Sukhija et al., 2024; Treven et al., 2024) and Gaussian process (GP) bandit optimization (Srinivas et al., 2012; Chowdhury & Gopalan, 2017; Scarlett et al., 2017) and is sublinear for common kernel such as the exponential kernel. To the best of our knowledge, we are the first to obtain regret bounds for the setting.
3. We evaluate NeoRL on several RL benchmarks against common model-based RL baselines. Our experimental results demonstrate that NeoRL consistently achieves sublinear regret, also when neural networks are employed instead of GPs for modeling dynamics. Moreover, in all our experiments, NeoRL converges to the optimal average cost.

# 2 Problem Setting

We consider a discrete-time dynamical system with running costs $c$.

$$\boldsymbol{x}_{t+1} = \boldsymbol{f}^*(\boldsymbol{x}_t, \boldsymbol{u}_t) + \boldsymbol{w}_t,\ (\boldsymbol{x}_t, \boldsymbol{u}_t) \in \mathcal{X} \times \mathcal{U},\ \boldsymbol{x}(0) = \boldsymbol{x}_0 \tag{1}$$

$$c(\boldsymbol{x}, \boldsymbol{u}) \in \mathbb{R}_{\geq 0} \tag{Running cost}$$

Here $\boldsymbol{x}_t \in \mathcal{X} \subseteq \mathbb{R}^{d_{\boldsymbol{x}}}$ is the state, $\boldsymbol{u}_t \in \mathcal{U} \subseteq \mathbb{R}^{d_{\boldsymbol{u}}}$ the control input, and $\boldsymbol{w}_t \in \mathcal{W} \subseteq \mathbb{R}^{\boldsymbol{w}}$ the process noise. The dynamics $\boldsymbol{f}^*$ are unknown and the cost $c$ is assumed to be known.

**Task** In this work, we study the average cost RL problem (Puterman, 2014), i.e., we want to learn the solution to the following minimization problem

$$A(\boldsymbol{\pi}^*, \boldsymbol{x}_0) = \min_{\boldsymbol{\pi} \in \Pi} A(\boldsymbol{\pi}, \boldsymbol{x}_0) = \min_{\boldsymbol{\pi} \in \Pi} \limsup_{T \to \infty} \frac{1}{T} \mathbb{E}_{\boldsymbol{\pi}} \left[ \sum_{t=0}^{T-1} c(\boldsymbol{x}_t, \boldsymbol{u}_t) \right]. \tag{2}$$

Moreover, we consider the nonepisodic RL setting where the system starts at an initial state $\boldsymbol{x}_0 \in \mathcal{X}$ but never resets back during learning, that is, we seek to learn online from a single trajectory. After each step $t$ in the environment, the RL system receives a transition tuple $(\boldsymbol{x}_t, \boldsymbol{u}_t, \boldsymbol{x}_{t+1})$ and updates its policy based on the data $\mathcal{D}_t$ collected thus far during learning. The average cost formulation is common for the nonepisodic setting (Jaksch et al., 2010; Abbasi-Yadkori & Szepesvári, 2011; Cohen et al., 2019; Dean et al., 2020; Simchowitz & Foster, 2020), and the cumulative regret for the learning algorithm in this case is defined as

$$R_T = \sum_{t=0}^{T-1} \mathbb{E}_{\boldsymbol{x}_t, \boldsymbol{u}_t | \boldsymbol{x}_0} [c(\boldsymbol{x}_t, \boldsymbol{u}_t) - A(\boldsymbol{\pi}^*, \boldsymbol{x}_0)]. \tag{3}$$

Studying the average cost criterion for general continuous state-action spaces is challenging even when the dynamics are known, since the average cost exists only for special classes of nonlinear systems (Arapostathis et al., 1993). In the following, we impose assumptions on the dynamics and policy class $\Pi$ that enable our theoretical analysis.

## 2.1 Assumptions

Imposing continuity on $\boldsymbol{f}^*$ is quite common in the control theory (Khalil, 2015) and reinforcement learning literature (Curi et al., 2020; Sussex et al., 2023; Sukhija et al., 2024). To this end, for our analysis, we make the following assumption.

**Assumption 2.1** (Continuity of $\boldsymbol{f}^*$ and $\boldsymbol{\pi}$)**.** The dynamics model $\boldsymbol{f}^*$ and all $\boldsymbol{\pi} \in \Pi$ are continuous.

Next, we make an assumption on the system's stochastic disturbances.

**Assumption 2.2** (Process noise distribution)**.** The process noise is i.i.d. Gaussian with variance $\sigma^2$, i.e., $\boldsymbol{w}_t \overset{i.i.d}{\sim} \mathcal{N}(\boldsymbol{0}, \sigma^2 \boldsymbol{I})$.

Our analysis can be extended for the more general heteroscedastic case, where $\sigma$ depends on $\boldsymbol{x}$. However, for simplicity, we focus on the homoscedastic setting. In the following, we make assumptions on our policy class. To this end, we first introduce the class of $\mathcal{K}_\infty$ functions.

**Definition 2.3** ($\mathcal{K}_\infty$-functions)**.** The function $\xi : \mathbb{R}_{\geq 0} \to \mathbb{R}_{\geq 0}$ is of class $\mathcal{K}_\infty$, if it is continuous, strictly increasing, $\xi(0) = 0$ and $\xi(s) \to \infty$ for $s \to \infty$.

**Assumption 2.4** (Policies with bounded energy)**.** We assume there exists $\kappa, \xi \in \mathcal{K}_\infty$, positive constants $K, C_u, C_l$ with $C_u > C_l$, and $\gamma \in (0, 1)$ such that for each $\boldsymbol{\pi} \in \Pi$ we have,

*Bounded energy:* There exists a Lyapunov function $V^{\boldsymbol{\pi}} : \mathcal{X} \to [0, \infty)$ for which $\forall \boldsymbol{x}, \boldsymbol{x}' \in \mathcal{X}$,

$$|V^{\boldsymbol{\pi}}(\boldsymbol{x}) - V^{\boldsymbol{\pi}}(\boldsymbol{x}')| \leq \kappa(\|\boldsymbol{x} - \boldsymbol{x}'\|) \qquad \text{(uniform continuity)}$$

$$C_l \xi(\|\boldsymbol{x}\|) \leq V^{\boldsymbol{\pi}}(\boldsymbol{x}) \leq C_u \xi(\|\boldsymbol{x}\|) \qquad \text{(positive definiteness)}$$

$$\mathbb{E}_{\boldsymbol{x}_+|\boldsymbol{x},\boldsymbol{\pi}}[V^{\boldsymbol{\pi}}(\boldsymbol{x}_+)] \leq \gamma V^{\boldsymbol{\pi}}(\boldsymbol{x}) + K \qquad \text{(drift condition)}$$

where $\boldsymbol{x}_+ = \boldsymbol{f}^*(\boldsymbol{x}, \boldsymbol{\pi}(\boldsymbol{x})) + \boldsymbol{w}$.

*Bounded norm of cost:*

$$\sup_{\boldsymbol{x} \in \mathcal{X}} \frac{c(\boldsymbol{x}, \boldsymbol{\pi}(\boldsymbol{x}))}{1 + V^{\boldsymbol{\pi}}(\boldsymbol{x})} < \infty$$

*Boundedness of the noise with respect to $\kappa$:*

$$\mathbb{E}_{\boldsymbol{w}}\left[\kappa(\|\boldsymbol{w}\|)\right] < \infty, \ \mathbb{E}_{\boldsymbol{w}}\left[\kappa^2(\|\boldsymbol{w}\|)\right] < \infty$$

The drift condition states that the energy between two timesteps can increase at most by $K$. In particular, the Lyapunov function $V^{\boldsymbol{\pi}}$ can be viewed as an energy function for the dynamical system, and the bounded energy condition above ensures that the system is not "blowing up". We do not perceive this as restrictive for real-world engineered systems. Other works that study learning nonlinear dynamics (Foster et al., 2020; Sattar & Oymak, 2022; Lale et al., 2021) in the nonepisodic setting also make stability assumptions such as global exponential stability for their analysis. In similar spirit, we make the bounded energy assumption for our policy class. The drift condition on the Lyapunov function is also used to study the ergodicity of Markov chains for continuous state spaces (Meyn & Tweedie, 2012; Hairer & Mattingly, 2011), which is crucial for our analysis of the infinite horizon behavior of the system. Moreover, for a very rich class of problems, the drift condition is satisfied. We highlight this in the corollary below.

**Lemma 2.5.** *Assume $\boldsymbol{f}^*$ is uniformly continuous and for all $\boldsymbol{\pi} \in \Pi$, $\boldsymbol{x} \in \mathcal{X}$, $\|\boldsymbol{\pi}(\boldsymbol{x})\| \leq u_{\max}$. Further assume, there exists $\boldsymbol{\pi}_s \in \Pi$ such that we have constants $K, C_u, C_l$ with $C_u > C_l$, $\gamma \in (0, 1)$, $\kappa, \alpha \in \mathcal{K}_\infty$ and a Lyapunov function $V : \mathcal{X} \to [0, \infty)$ for which $\forall \boldsymbol{x}, \boldsymbol{x}' \in \mathcal{X}$,*

$$|V(\boldsymbol{x}) - V(\boldsymbol{x}')| \leq \kappa(\|\boldsymbol{x} - \boldsymbol{x}'\|)$$

$$C_l \xi(\|\boldsymbol{x}\|) \leq V(\boldsymbol{x}) \leq C_u \xi(\|\boldsymbol{x}\|)$$

$$\mathbb{E}_{\boldsymbol{x}_+|\boldsymbol{x},\boldsymbol{\pi}_s}[V(\boldsymbol{x}_+)] \leq \gamma V(\boldsymbol{x}) + K,$$

*where $\boldsymbol{x}_+ = \boldsymbol{f}^*(\boldsymbol{x}, \boldsymbol{\pi}(\boldsymbol{x})) + \boldsymbol{w}$. Then, $V$ also satisfies the drift condition for all $\boldsymbol{\pi} \in \Pi$, i.e., is a Lyapunov function for all policies.*

We prove this lemma in Appendix A. Intuitively, if the inputs are bounded, the energy inserted into the system by another policy is also bounded. Nearly all real-world systems have bounded inputs due to the physical limitations of actuators. For these systems, it suffices if only one policy in $\Pi$ satisfies the drift condition.

The boundedness assumptions for the cost and the noise in Assumption 2.4 are satisfied for a rich class of cost and $\mathcal{K}_\infty$ functions.

Under these assumptions, we can show the existence of the average cost solution.

**Theorem 2.6** (Existence of Average Cost Solution)**.** *Let Assumption 2.1 – 2.4 hold. Consider any $\boldsymbol{\pi} \in \Pi$ and let $P^{\boldsymbol{\pi}}$ denote its transition kernel, i.e., $P^{\boldsymbol{\pi}}(\boldsymbol{x}, \mathcal{A}) = \mathbb{P}(\boldsymbol{x}_+ \in \mathcal{A}|\boldsymbol{x}, \boldsymbol{\pi}(\boldsymbol{x}))$ for $\mathcal{A} \subseteq \mathcal{X}$. Then $P^{\boldsymbol{\pi}}$ admits a unique invariant measure $\bar{P}^{\boldsymbol{\pi}}$, and there exists $C_2, C_3 \in (0, \infty)$, $\lambda \in (0, 1)$ such that*

Average Cost*;*

$$A(\boldsymbol{\pi}) = \lim_{T \to \infty} \frac{1}{T} \mathbb{E}_{\boldsymbol{\pi}}\left[\sum_{t=0}^{T-1} c(\boldsymbol{x}_t, \boldsymbol{u}_t)\right] = \mathbb{E}_{\boldsymbol{x} \sim \bar{P}^{\boldsymbol{\pi}}}\left[c(\boldsymbol{x}, \boldsymbol{\pi}(x))\right]$$

Bias Cost; *Letting* $B(\boldsymbol{\pi}, \boldsymbol{x}_0) = \lim_{T\to\infty} \mathbb{E}_{\boldsymbol{\pi}} \left[\sum_{t=0}^{T-1} c(\boldsymbol{x}_t, \boldsymbol{u}_t) - A(\boldsymbol{\pi})\right]$ *denote the bias, we have*

$$|B(\boldsymbol{\pi}, \boldsymbol{x}_0)| \leq C_2(1 + V^{\boldsymbol{\pi}}(\boldsymbol{x}_0))\frac{1}{1-\lambda}$$

*for all* $\boldsymbol{x}_0 \in \mathcal{X}$.

Theorem 2.6 is a crucial result for our analysis since it implies that the average cost is bounded and *independent of the initial state* $\boldsymbol{x}_0$. Furthermore, it also shows that the bias is bounded. The average cost criterion satisfies the following Bellman equation (Puterman, 2014) below

$$B(\boldsymbol{\pi}, \boldsymbol{x}) + A(\boldsymbol{\pi}) = c(\boldsymbol{x}, \boldsymbol{\pi}(\boldsymbol{x})) + \mathbb{E}_{\boldsymbol{x}_+}[B(\boldsymbol{\pi}, \boldsymbol{x}_+)|\boldsymbol{x}, \boldsymbol{\pi}] \tag{4}$$

Accordingly, the bias term plays an important role in the regret analysis (also notice its similarity to our regret term in Equation (3)).

Thus far, we have only made assumptions that make the average cost problem tractable. In the following, we make an assumption on the dynamics that allow us to learn it from data. Moreover, we assume that at each step $n$ we learn a mean estimate $\boldsymbol{\mu}_n$ of $\boldsymbol{f}^*$ and can quantify our uncertainty $\boldsymbol{\sigma}_n$ over the estimate. More formally, we learn a well-calibrated statistical model of $\boldsymbol{f}^*$ as defined below.

**Definition 2.7** (Well-calibrated statistical model of $\boldsymbol{f}^*$, Rothfuss et al. (2023))**.** Let $\mathcal{Z} \stackrel{\text{def}}{=} \mathcal{X} \times \mathcal{U}$. An all-time well-calibrated statistical model of the function $\boldsymbol{f}^*$ is a sequence $\{\mathcal{M}_n(\delta)\}_{n\geq 0}$, where

$$\mathcal{M}_n(\delta) \stackrel{\text{def}}{=} \left\{\boldsymbol{f} : \mathcal{Z} \to \mathbb{R}^{d_x} \mid \forall \boldsymbol{z} \in \mathcal{Z}, \forall j \in 1, \dots, d_x : |\mu_{n,j}(\boldsymbol{z}) - f_j(\boldsymbol{z})| \leq \beta_n(\delta)\sigma_{n,j}(\boldsymbol{z})\right\},$$

if, with probability at least $1-\delta$, we have $\boldsymbol{f}^* \in \bigcap_{n\geq 0} \mathcal{M}_n(\delta)$. Here, $f_j$, $\mu_{n,j}$ and $\sigma_{n,j}$ denote the $j$-th element in the vector-valued functions $\boldsymbol{f}$, $\boldsymbol{\mu}_n$ and $\boldsymbol{\sigma}_n$ respectively, and $\beta_n(\delta) \in \mathbb{R}_{\geq 0}$ is a scalar function that depends on the confidence level $\delta \in (0, 1]$ and which is monotonically increasing in $n$.

Next, we assume that $\boldsymbol{f}^*$ resides in a Reproducing Kernel Hilbert Space (RKHS) of vector-valued functions and show that this is sufficient for us to obtain a well-calibrated model.

**Assumption 2.8.** We assume that the functions $f_j^*$, $j \in 1, \dots, d_x$ lie in a RKHS with kernel $k$ and have a bounded norm $B$, that is $\boldsymbol{f}^* \in \mathcal{H}_{k,B}^{d_x}$, with $\mathcal{H}_{k,B}^{d_x} = \{\boldsymbol{f} \mid \|f_j\|_k \leq B, j = 1, \dots, d_x\}$. Moreover, we assume that $k(\boldsymbol{x}, \boldsymbol{x}) \leq \sigma_{\max}$ for all $\boldsymbol{x} \in \mathcal{X}$.

Assumption 2.8 allows us to model $\boldsymbol{f}^*$ with GPs for which the mean and epistemic uncertainty ($\boldsymbol{\mu}_n(\boldsymbol{z}) = [\mu_{n,j}(\boldsymbol{z})]_{j\leq d_x}$, and $\boldsymbol{\sigma}_n(\boldsymbol{z}) = [\sigma_{n,j}(\boldsymbol{z})]_{j\leq d_x}$) have an analytical formula

$$\begin{aligned} \mu_{n,j}(\boldsymbol{z}) &= \boldsymbol{k}_n^\top(\boldsymbol{z})(\boldsymbol{K}_n + \sigma^2\boldsymbol{I})^{-1}\boldsymbol{y}_{1:n}^j, \\ \sigma_{n,j}^2(\boldsymbol{z}) &= k(\boldsymbol{x}, \boldsymbol{x}) - \boldsymbol{k}_n^\top(\boldsymbol{z})(\boldsymbol{K}_n + \sigma^2\boldsymbol{I})^{-1}\boldsymbol{k}_n(\boldsymbol{x}), \end{aligned} \tag{5}$$

Here, $\boldsymbol{y}_{1:n}^j$ corresponds to the noisy measurements of $f_j^*$, i.e., the observed next state from the transitions dataset $\mathcal{D}_{1:n}$, $\boldsymbol{k}_n = [k(\boldsymbol{z}, \boldsymbol{z}_i)]_{i\leq nT}$, $\boldsymbol{z}_i \in \mathcal{D}_{1:n}$, and $\boldsymbol{K}_n = [k(\boldsymbol{z}_i, \boldsymbol{z}_l)]_{i,l\leq nT}$, $\boldsymbol{z}_i, \boldsymbol{z}_l \in \mathcal{D}_{1:n}$ is the data kernel matrix. The restriction on the kernel $k(\boldsymbol{x}, \boldsymbol{x}) \leq \sigma_{\max}$ implies boundedness of $\boldsymbol{f}^*$ and has also appeared in works studying the episodic setting for nonlinear systems (Mania et al., 2020; Kakade et al., 2020; Curi et al., 2020; Sukhija et al., 2024; Wagenmaker et al., 2023). We can also define $\boldsymbol{f}^*$ such that $\boldsymbol{x}_k = \boldsymbol{x}_{k-1} + \boldsymbol{f}^*(\boldsymbol{x}_{k-1}, \boldsymbol{u}_{k-1}) + \boldsymbol{w}_{k-1}$ in which case the boundedness of $\boldsymbol{f}^*$ captures many real-world systems.

**Lemma 2.9** (Well calibrated confidence intervals for RKHS, Rothfuss et al. (2023))**.** *Let* $\boldsymbol{f}^* \in \mathcal{H}_{k,B}^{d_x}$. *Suppose* $\boldsymbol{\mu}_n$ *and* $\boldsymbol{\sigma}_n$ *are the posterior mean and variance of a GP with kernel* $k$*, c.f., Equation* (5)*. There exists* $\beta_n(\delta) \propto \sqrt{\Gamma_n}$*, for which the tuple* $(\boldsymbol{\mu}_n, \boldsymbol{\sigma}_n, \beta_n(\delta))$ *is a well-calibrated statistical model of* $\boldsymbol{f}^*$.

In summary, in the RKHS setting, a GP is a well-calibrated model. For more general models like Bayesian neural networks (BNNs), methods such as Kuleshov et al. (2018) can be used for calibration. Our results can also be extended beyond the RKHS setting to other classes of well-calibrated models similar to Curi et al. (2020).

**Algorithm 1 NEORL:** NONEPISODIC OPTIMISTIC RL

**Init:** Aleatoric uncertainty $\sigma$, Probability $\delta$, Statistical model $(\boldsymbol{\mu}_0, \boldsymbol{\sigma}_0, \beta_0(\delta))$, $H_0$

**for** $n = 1, \ldots, N$ **do**

$\boldsymbol{\pi}_n = \arg\min_{\boldsymbol{\pi}\in\Pi} \min_{\boldsymbol{f}\in\mathcal{M}_{n-1}\cap\mathcal{M}_0} A(\boldsymbol{\pi}, \boldsymbol{f})$ ➤ Prepare policy

$H_n = 2H_{n-1}$ ➤ Set horizon

$\mathcal{D}_n \leftarrow \text{ROLLOUT}(\boldsymbol{\pi}_n)$ ➤ Collect measurements for horizon $H_n$

Update $(\boldsymbol{\mu}_n, \boldsymbol{\sigma}_n, \beta_n) \leftarrow \mathcal{D}_n$ ➤ Update statistical model $\mathcal{M}_n$

**end for**

## 3 NEORL

In the following, we present our algorithm: **N**on**e**pisodic **O**ptimistic **RL** (NEORL) for efficient nonepisodic exploration in continuous state-action spaces. NEORL builds on recent advances in episodic RL (Kakade et al., 2020; Curi et al., 2020; Sukhija et al., 2024; Treven et al., 2024) and leverages the optimism in the face of uncertainty paradigm to pick policies that are optimistic w.r.t. the dynamics within our calibrated statistical model as follows

$$(\boldsymbol{\pi}_n, \boldsymbol{f}_n) \stackrel{\text{def}}{=} \arg\min_{\boldsymbol{\pi}\in\Pi,\ \boldsymbol{f}\in\mathcal{M}_{n-1}\cap\mathcal{M}_0} A(\boldsymbol{\pi}, \boldsymbol{f}). \tag{6}$$

Here, $\boldsymbol{f}_n$ is a dynamical system such that the cost by controlling $\boldsymbol{f}_n$ with its optimal policy $\boldsymbol{\pi}_n$ is the lowest among all the plausible systems from $\mathcal{M}_{n-1} \cap \mathcal{M}_0$. Note, from Lemma 2.9 we have that $\boldsymbol{f}^* \in \mathcal{M}_{n-1} \cap \mathcal{M}_0$ (with high probability) and therefore the solution to Equation (6) gives an optimistic estimate for the average cost. We take the intersection of $\mathcal{M}_{n-1}$ with $\mathcal{M}_0$ to ensure that we maintain at least the same confidence about our model as at the beginning, i.e., $n = 0$, during learning.

NEORL proceeds in the following manner. Similar to Jaksch et al. (2010), we bin the total time $T$ the agent spends interacting in the environment into $N$ "artificial" episodes. At each episode, we pick a policy according to Equation (6) and roll it out for $H_n$ steps on the system. Next, we use the data collected during the rollout to update our statistical model. Finally, we double the horizon $H_{n+1} = 2H_n$, akin to Simchowitz & Foster (2020), and continue to the next episode *without resetting* the system back to the initial state $\boldsymbol{x}_0$. Intuitively, in the beginning, when our model estimate is not accurate, we update our model more frequently, and with more episodes as our model gets better we reduce the frequency of updates. The algorithm is summarized in Algorithm 1.

### 3.1 Theoretical Results

In the following, we study the theoretical properties for NEORL and provide a first-of-its-kind bound on the cumulative regret for the average cost criterion for general nonlinear dynamical systems. Our bound depends on the *maximum information gain* of kernel $k$ (Srinivas et al., 2012), defined as

$$\Gamma_T(k) = \max_{\mathcal{A}\subset\mathcal{X}\times\mathcal{U};|\mathcal{A}|\leq T} \frac{1}{2}\log\left|\boldsymbol{I} + \sigma^{-2}\boldsymbol{K}_T\right|.$$

$\Gamma_T$ represents the complexity of learning $\boldsymbol{f}^*$ from $T$ data points and is sublinear for a very rich class of kernels (e.g., $\mathcal{O}(\log^{d_x+d_u+1}(T))$ for the exponential (RBF) kernel, $\mathcal{O}((d_x + d_u)\log(T))$ for the linear kernel). In Appendix A, we report the dependence of $\Gamma_T$ on $T$ in Table 1.

**Theorem 3.1** (Cumulative Regret of NEORL)**.** *Let Assumption 2.1 – 2.8 hold, and define $H_0$ as the smallest integer such that*

$$H_0 > \frac{\log\left({C_u}/{C_l}\right)}{\log\left({1}/{\gamma}\right)}.$$

*Then with probability at least $1 - \delta$, we have the following regret for* NEORL

$$R_T \leq D_4(\boldsymbol{x}_0, K, \gamma)\beta_T\sqrt{T\Gamma_T} + D_5(\boldsymbol{x}_0, K, \gamma)\log_2\left(\frac{T}{H_0} + 1\right). \tag{7}$$

*with $D_4(\boldsymbol{x}_0, K, \gamma)$, $D_5(\boldsymbol{x}_0, K, \gamma)$ being bounded constants for bounded $\|\boldsymbol{x}_0\|$, $K$, and $\gamma < 1$.*

From Lemma 2.9 we have that $\beta_T \propto \sqrt{\Gamma_T}$ and therefore Theorem 3.1 gives sublinear regret for a rich class of RKHS functions. Moreover, it also gives a minimal horizon $H_0$ that we need to maintain before switching to the next policy. Even for the linear case, fast switching between stable controllers can destabilize the closed-loop system. We ensure this does not happen in our case by having a minimal horizon of $H_0$. Theorem 3.1 can also be derived beyond the RKHS setting for a more general class of well-calibrated models. In this case, the maximum information gain is replaced by the model complexity from Curi et al. (2020) (c.f., Curi et al. (2020); Sukhija et al. (2024) for further detail).

In the following, we give an intuitive proof sketch for Theorem 3.1. The detailed proof is provided in Appendix A.

**Proof sketch** The proof can be split into three main steps. First, we show the ergodicity of the closed-loop system, a sufficient condition for showing the existence of the average cost and bias term, i.e., Theorem 2.6, for every policy $\boldsymbol{\pi} \in \Pi$ under Assumption 2.1 – 2.4. For this, we use elementary results on Markov chains in measurable spaces from Meyn & Tweedie (2012); Hairer & Mattingly (2011). Second, we show that under Assumption 2.8, the optimistic system selected in Equation (6), retains the same properties as the true system $\boldsymbol{f}^*$, e.g., stability, and therefore also is ergodic. Crucial to show this is that the true system $\boldsymbol{f}^*$ and the optimistic system $\boldsymbol{f}_n$ are at most $\beta_n \boldsymbol{\sigma}_n$ apart. Finally, in the third step, we show that as we update our model and policy every $H_n$ steps, the doubling of the horizon retains the system properties from above, and our accumulated model uncertainties across $T$ environment steps grow with the rate $\Gamma_T$. For the latter, we use the analysis from Kakade et al. (2020) for the episodic case, to bound the deviation between the optimistic average cost and the true average cost.

### 3.2 Practical Modifications

For testing NeoRL, we make three modifications that simplify its deployment in practice in terms of implementation and computation time. First, instead of doubling the horizon $H_n$ we pick a fixed horizon $H$ during the experiment. This makes the planning and training of the agent easier. Next, we use a receding horizon controller, i.e., model predictive control (MPC) (García et al., 1989), instead of directly optimizing for the average cost in Equation (6). MPC is widely used to obtain a feedback controller for the infinite horizon setting. Moreover, while for linear systems, the Riccati equations (Anderson & Moore, 2007) provide an analytical solution to Equation (2), no such solution exists for the nonlinear case and MPC is commonly used as an approximation. Further, under additional assumptions on the cost and dynamics, MPC also obtains a policy with bounded average cost, which is crucial for the nonepisodic case (c.f., Assumption 2.4). We use the iCEM optimizer for planning (Pinneri et al., 2021). Finally, instead of optimizing over $\mathcal{M}_n \cap \mathcal{M}_0$, we optimize directly over $\mathcal{M}_n$. This allows us to use the reparameterization trick from Curi et al. (2020) and obtain a simple and tractable optimization problem. In summary, for each step $t$ in the environment, we solve the following optimization problem

$$\min_{\boldsymbol{u}_{0:H_{\text{MPC}}-1}, \boldsymbol{\eta}_{0:H_{\text{MPC}}-1}} \mathbb{E}\left[\sum_{h=0}^{H_{\text{MPC}}-1} c(\hat{\boldsymbol{x}}_h, \boldsymbol{u}_h)\right], \tag{8}$$
$$\text{s.t. } \hat{\boldsymbol{x}}_{h+1} = \boldsymbol{\mu}_{n-1}(\hat{\boldsymbol{x}}_h, \boldsymbol{u}_h) + \beta_{n-1}(\delta)\boldsymbol{\sigma}_{n-1}(\hat{\boldsymbol{x}}_h, \boldsymbol{u}_h)\boldsymbol{\eta}_h + \boldsymbol{w}_h \text{ and } \hat{\boldsymbol{x}}_0 = \boldsymbol{x}_t.$$

Here $H_{\text{MPC}}$ is the MPC horizon. We take the first input from the solution of the problem above, i.e., $\boldsymbol{u}_0^*$, and execute this in the system. We then repeat this procedure for $H$ steps and then update our statistical model $\mathcal{M}_n$. The resulting optimization above considers a larger action space as it includes the hallucinated controls $\boldsymbol{\eta}$ as additional input variables. The hallucinated controls are introduced through the reparameterization trick from (Curi et al., 2020) and are used to directly optimize over models in $\boldsymbol{f} \in \mathcal{M}_{n-1}$. Moreover, the final algorithm can be seen as a natural extension to H-UCRL (Curi et al., 2020) for the nonepisodic setting. We summarize the algorithm in Appendix B Algorithm 2. Note while these modifications deviate from our theoretical analysis, empirically they work well for GP and BNN models, c.f., Section 4.

## 4 Experiments

We evaluate NeoRL on the Pendulum-v1 and MountainCar environment from the OpenAI gym benchmark suite (Brockman et al., 2016), Cartpole, Reacher, and Swimmer from the DeepMind control suite (Tassa et al., 2018), the racecar simulator from Kabzan et al. (2020), and a soft robotic arm from Tekinalp et al. (2024). The swimmer and the soft robotic arm are fairly high-dimensional systems – the swimmer has a 28-dimensional state and 5-dimensional action space, and the soft arm

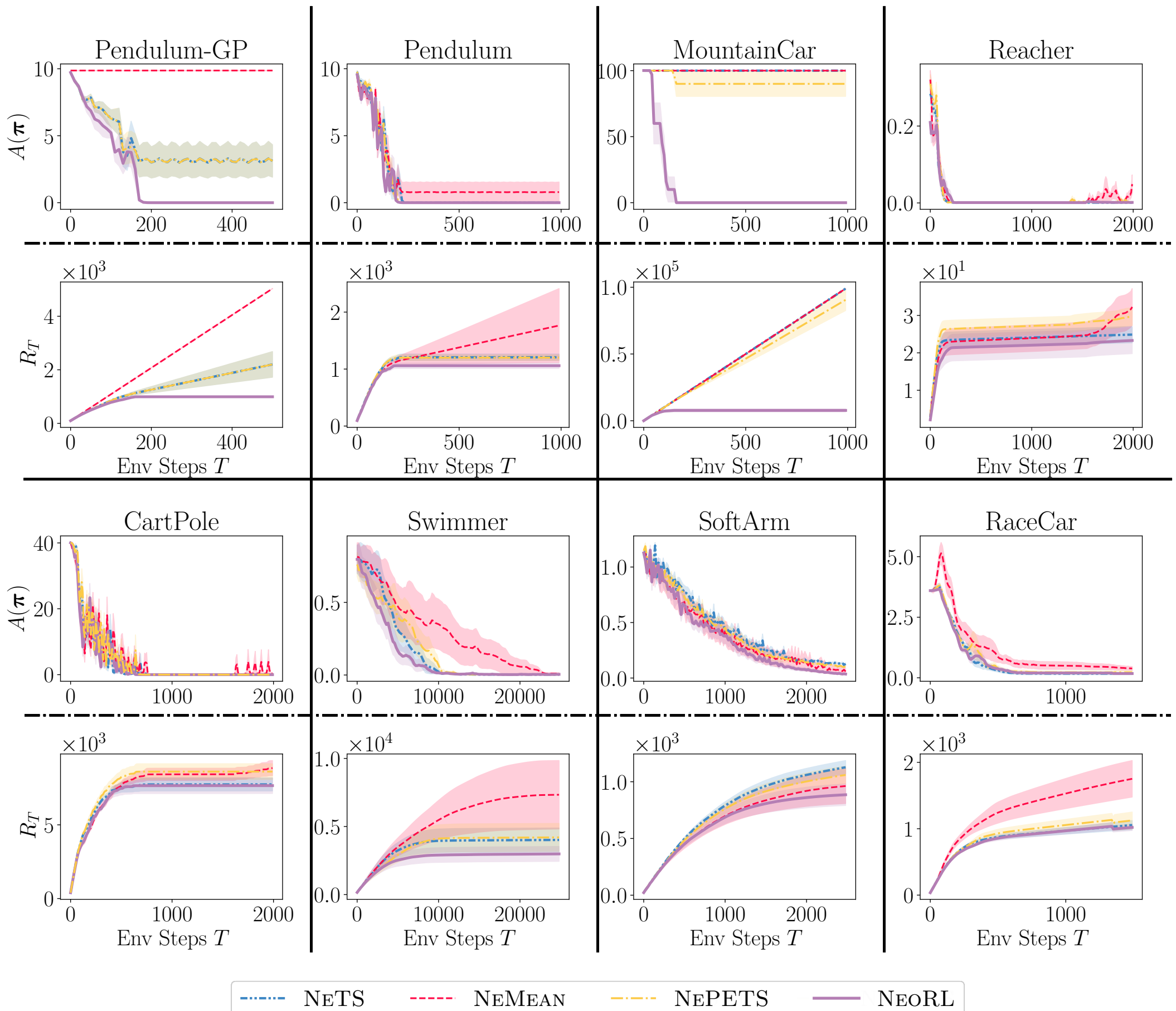

Figure 1: Average reward $A(\boldsymbol{\pi})$ and cumulative regret $R_T$ over ten different seeds for all environments. We report the mean performance with one standard error as shaded regions. During all experiments, the environment is never reset. For all baselines, we model the dynamics with probabilistic ensembles, except in the Pendulum-GP experiment, where GPs are used instead. NeoRL significantly outperforms all baselines and converges to the optimal average reward, $A(\boldsymbol{\pi}^*) = 0$, showing sublinear cumulative regret $R_T$ for all environments.

is represented by a 58-dimensional state and has a 12-dimensional action space. All environments are never reset during learning. Moreover, the Pendulum-v1, MountainCar, CartPole, and Reacher environments operate within a bounded domain and thus inherently satisfy Assumption 2.4. The swimmer, racecar, and soft arm can operate in an unbounded domain but have a cost function that penalizes the distance between the system's state $\boldsymbol{x}_t$ and a target state $\boldsymbol{x}^*$. Therefore, the cost encourages the system to move towards the target and remain within a bounded domain.

**Baselines** In the episodic setting, resets can be used to control the exploration space for the agent. However, in the absence of resets, the agent can explore arbitrarily and end up in states that are irrelevant to the task at hand. Moreover, the agent has to follow an uninterrupted chain of experience, which makes the nonepisodic setting the most challenging one in RL (Kakade, 2003). Accordingly, there are only a few algorithms that consider this setting (c.f., Section 5). In this work, we focus on model-based RL (MBRL) algorithms due to their sample efficiency. In particular, we adopt common MBRL methods for our setting. MBRL algorithms typically differentiate in three ways; (*i*) propagating dynamics for planning (Chua et al., 2018; Osband & Van Roy, 2017; Kakade et al., 2020; Curi et al., 2020), (*ii*) representation of the dynamics model (Ha & Schmidhuber, 2018; Hafner et al., 2019; Kipf et al., 2019), and (*iii*) types of planners (Williams et al., 2017; Hafner et al., 2020; Pinneri et al., 2021). NeoRL is independent to the choice of representation or planners. Therefore, we focus on (*i*) and use probabilistic ensembles (Lakshminarayanan et al., 2017) and GPs for modeling our dynamics and MPC with iCEM (Pinneri et al., 2021) as the planner. Common techniques to propagate the dynamics for planning are using the mean, trajectory sampling (Chua et al., 2018), and Thompson

sampling (Osband & Van Roy, 2017). We adapt these three for our setting similar to as discussed in Section 3.2. For all experiments with probabilistic ensembles, we consider TS1 from Chua et al. (2018) for trajectory sampling, and for the GP experiment, we use distribution sampling from Chua et al. (2018). We call the three baselines NEMEAN (nonepisodic mean), NEPETS (nonepisodic PETS), and NETS (nonepisodic Thompson sampling). NEMEAN and NEPETS are greedy w.r.t. the current estimate of the dynamics, i.e., do not explicitly encourage exploration. In our experiments, we show that being greedy does not suffice to converge to the optimal average cost, that is, obtain sublinear regret. The code for our experiments is available online.[2]

**Convergence to the optimal average cost** In Figure 1 we report the normalized average cost and cumulative regret of NEORL, NEMEAN, NEPETS, and NETS. The normalized average cost is defined such that $A(\boldsymbol{\pi}^*) = 0$ for all environments. We observe that NEMEAN fails to converge to the optimal average cost for the Pendulum-v1 environment for both probabilistic ensembles and a GP model. It also fails to solve the MountainCar environment and is unstable for the Reacher and CartPole. In general, NEMEAN performs the worst among all methods. This is similar to the episodic case, where using the mean model often leads to the policy "overfitting" to the model inaccuracies (Chua et al., 2018). NEPETS performs better than the mean, however still significantly worse than NEORL. Even in the episodic setting, PETS tends to underexplore (Curi et al., 2020). We observe the same for the nonepisodic case, especially for the MountainCar task, which is a challenging RL environment with a sparse cost. Here NEPETS is also not able to achieve the optimal average cost and thus does not have sublinear cumulative regret. NETS performs similarly to NEPETS and is also not able to solve the MountainCar task.

NEORL performs the best among the baselines for all experiments and converges to the optimal average cost achieving sublinear cumulative regret using only $\sim 10^3$ environment interactions. Moreover, this observation is consistent between different dynamics models (GPs and probabilistic ensembles) and environments. Even in environments that are unbounded, i.e., Swimmer, SoftArm, and RaceCar, we observe that NEORL converges to the optimal average cost the fastest. We believe this is due to the feedback control from MPC, which has a stabilizing effect.

**Calling reset when needed** All the experiments in Figure 1 considered the nonepisodic setting where the system was never reset during learning. A special case of our theoretical analysis is the class of policies $\Pi$ that may call for a reset / "ask for help" whenever they end up in an undesirable part of the state space. In this setting, the system is typically restricted to a compact subset of the state space $\mathcal{X}$, and the policy class satisfies Assumption 2.4. For many real-world applications, such a policy class can be derived. To simulate this experiment, we consider the CartPoleBalance task in Figure 2, where the goal is to balance the pole in the upright position. A reset is triggered whenever the pole drops. We again observe that NEORL achieves the best performance, i.e., lowest cumulative regret and thus learns to solve the task the fastest. Moreover, it also requires fewer resets than NEMEAN, NEPETS, and NETS.

## 5 Related Work

**Average cost RL for finite state-action spaces** A significant amount of work studies the average cost/reward RL setting for finite-state action spaces. Moreover, seminal algorithms such as $E^3$ (Kearns & Singh, 2002) and $\text{R-max}$ (Brafman & Tennenholtz, 2002) have established PAC bounds for the nonepisodic setting. These bounds are further improved for communicating MDPs by the UCRL2 (Jaksch et al., 2010) algorithm, which, similar to NEORL, is based on the optimism in the face of uncertainty paradigm and picks policies that are optimistic w.r.t. to the estimated dynamics. Their result is extended for weakly-communicating MDPs by REGAL (Bartlett & Tewari, 2012), similar results are derived for Thompson sampling based exploration (Ouyang et al., 2017), and for factored-MDP (Xu & Tewari, 2020). Albeit the significant amount of work for the finite case, progress for continuous state-action spaces has mostly been limited to linear dynamical systems.

**Nonepisodic RL for linear systems** There is a large body of work for nonepisodic learning with linear systems (Abbasi-Yadkori & Szepesvári, 2011; Cohen et al., 2019; Simchowitz & Foster, 2020; Dean et al., 2020; Lale et al., 2020; Faradonbeh et al., 2020; Abeille & Lazaric, 2020; Treven et al., 2021). For linear systems with quadratic costs, the average reward problem, also known as the linear quadratic-Gaussian (LQG), has a closed-form solution which is obtained via the Riccati equations (Anderson & Moore, 2007). Moreover, for LQG, stability and optimality are intertwined,

[2] https://github.com/lasgroup/opax/tree/neorl

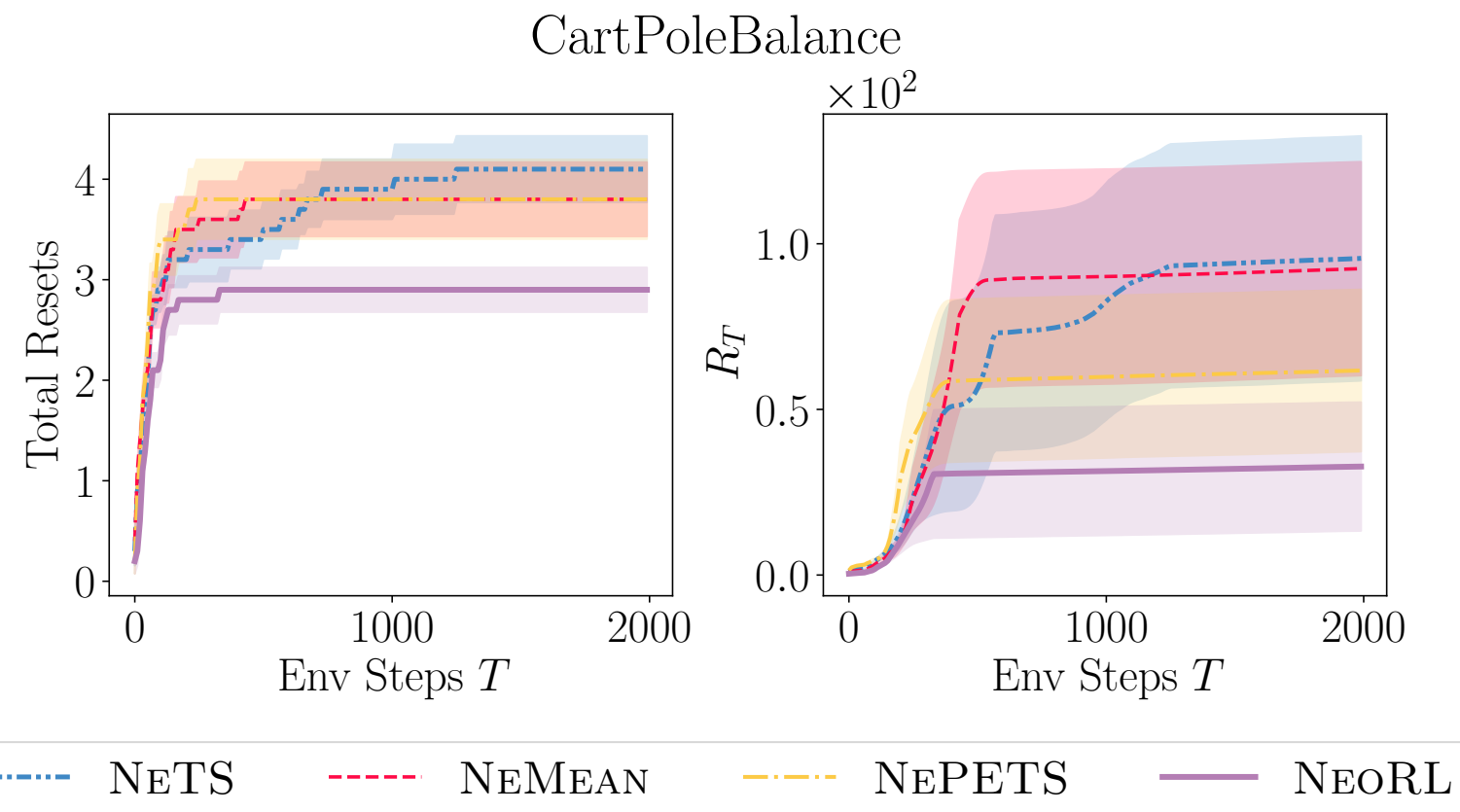

Figure 2: Total number of resets and cumulative regret $R_T$ for the cart pole balancing task over ten different seeds. We report the mean performance with one standard errors as the shaded region. The environment is automatically reset whenever the agent drops the pole. All baselines solve the task, but NeoRL converges the fastest requiring fewer resets and suffering smaller regret.

making studying linear systems much easier than their nonlinear counterpart. For studying nonlinear systems, additional assumptions on their stability are usually made.

**Episodic RL for nonlinear systems** In the case of nonlinear systems, guarantees have mostly been established for the episodic setting (Mania et al., 2020; Kakade et al., 2020; Curi et al., 2020; Wagenmaker et al., 2023; Sukhija et al., 2024; Treven et al., 2024). In this setting, the agent begins each episode from an initial state $s_0$ (or initial state distribution) and interacts with the environment for a fixed horizon $H$. It uses the data collected from the interactions to update its model. After each episode, the agent is reset back to $s_0$. The works mentioned above theoretically study this setting for finite-horizon MDPs and establish regret bounds for general nonlinear systems. Particularly Kakade et al. (2020); Curi et al. (2020); Sukhija et al. (2024); Treven et al. (2024) also use an optimism-based approach similar to ours. Compared to the nonepisodic case, the analysis of episodic RL methods is simpler as resets restrict the agent's exploration around the initial state $s_0$ and prevent the system from blowing up or visiting states from which the agent cannot recover. However, as discussed in Section 1, resets are often prohibitive and RL agents that learn non-episodically are preferred for many real-world applications.

**Nonepisodic RL beyond linear systems** Only a few works consider the nonepisodic/single-trajectory case. For instance, a line of work studies data-driven MPC approaches focusing mostly on establishing system-theoretic guarantees such as closed-loop stability and robustness (Berberich & Allgöwer, 2024). From the learning side, Foster et al. (2020); Sattar & Oymak (2022) study the problem of system identification of a closed-loop globally exponentially stable dynamical system from a single trajectory. Lale et al. (2021) study the nonepisodic setting for nonlinear systems with MPC. Moreover, they consider finite-order or exponentially fading NARX systems that lie in the RKHS of infinitely smooth functions, which they further approximate with random Fourier features (Rahimi & Recht, 2007) $\phi$ with feature size $D$. Further, they assume access to bounded persistently exciting inputs w.r.t. the feature matrix $\mathbf{\Phi}_t\mathbf{\Phi}_t^\intercal$. This assumption is generally tough to verify and common excitation strategies such as random exploration often don't perform well for nonlinear systems (Sukhija et al., 2024). The algorithm also operates in two stages, where in the first stage it performs pure exploration for system identification and in the second stage exploitation, i.e., acting greedily w.r.t. the estimated dynamics, akin to NeMean. Additionally, the algorithm requires the feature size $D$ to increase with the horizon $T$. They give a regret bound of $\mathcal{O}\left(T^{2/3}\right)$ where the regret is measured w.r.t. to the oracle MPC with access to the true dynamics. Lale et al. (2021) also assume exponential input-to-output stability of the system to avoid blow-up during exploration. Our work considers more general RKHS, naturally trades-off exploration and exploitation, does not require apriori knowledge of persistently exciting inputs and gives a regret bound of $\mathcal{O}(\beta_T\sqrt{T\Gamma_T})$ w.r.t. the optimal average cost criterion. Moreover, our regret bound is similar to the ones obtained for nonlinear systems in the episodic case and Gaussian process bandits (Srinivas et al., 2012; Chowdhury & Gopalan, 2017; Scarlett et al., 2017). To the best of our knowledge, we are the first to give such a regret bound for nonlinear systems.

**Nonepisodic Deep RL** Standard deep RL approaches often fail in the nonepisodic setting (Sharma et al., 2021b). To this end, deep RL algorithms have also been developed for the nonepisodic case. Mostly, these works focus on learning to reset and formulate it from the perspective of safety (Eysenbach et al., 2018) (avoiding undesirable states), chaining multiple controllers (Han et al., 2015), skill discovery/intrinsic exploration (Zhu et al., 2020; Xu et al., 2020), curriculum learning (Sharma et al., 2021a), and learning initial state distributions from demonstrations (Sharma et al., 2022). However, in contrast to us, none of the works above provide any theoretical guarantees.

There are several extensions of model-free deep RL algorithms to the average reward setting (TRPO (Zhang & Ross, 2021), PPO (Ma et al., 2021), and DDPG (Saxena et al., 2023)). However, they mostly focus on maximizing the long-term behavior of the RL agent and allow for resets during learning. Overall, extending RL algorithms for the discounted case to the average one is still an open problem (Dewanto et al., 2020). However, future work in this direction will benefit NEORL. Since average-reward optimizers can be used in combination with NEORL to directly minimize the average cost in a model-based policy optimization (Janner et al., 2019) manner.

## 6 Conclusion

We propose, NEORL, a novel model-based RL algorithm for the nonepisodic setting with nonlinear dynamics and continuous state and action spaces. NEORL seeks for average-cost optimal policies and leverages the model's epistemic uncertainty to perform optimistic exploration. Similar to the episodic case (Kakade et al., 2020; Curi et al., 2020), we provide a regret bound for NEORL of $\mathcal{O}(\beta_T\sqrt{T\Gamma_T})$ for Gaussian process dynamics. To our knowledge, we are the first to obtain this result in the nonepisodic setting. We compare NEORL to other model-based RL methods on standard deep RL benchmarks. Our experiments demonstrate that NEORL, converges to the optimal average cost of $A(\boldsymbol{\pi}^*) = 0$ across all environments, suffering sublinear regret even when Bayesian neural networks are used to model the dynamics. Moreover, NEORL outperforms all our baselines across all environments requiring only $\sim 10^3$ samples for learning.

Future work may consider deriving lower bounds on the regret of NEORL, studying different assumptions on $\boldsymbol{f}^*$ and $\Pi$, and investigating different notions of optimality such as bias optimality in the nonepisodic setting (Mahadevan, 1996).

## Acknowledgments and Disclosure of Funding

We would like to thank Mohammad Reza Karimi, Scott Sussex, and Armin Lederer for the insightful discussions and feedback on this work. This project has received funding from the Swiss National Science Foundation under NCCR Automation, grant agreement 51NF40 180545, and the Microsoft Swiss Joint Research Center.

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

# Appendices

## A Proofs

In this section, we prove Theorem 2.6 and Theorem 3.1. First, we start with the proof of Lemma 2.5.

*Proof of Lemma 2.5.* We first analyze the following term $\mathbb{E}_{\boldsymbol{w}}[V(\boldsymbol{f}^*(\boldsymbol{x},\boldsymbol{\pi}(\boldsymbol{x}))+\boldsymbol{w})-V(\boldsymbol{f}^*(\boldsymbol{x},\boldsymbol{\pi}_s(\boldsymbol{x}))+\boldsymbol{w})]$ for any $\boldsymbol{\pi}\in\Pi$.

$$
\begin{aligned}
&\mathbb{E}_{\boldsymbol{w}}[V(\boldsymbol{f}^*(\boldsymbol{x},\boldsymbol{\pi}(\boldsymbol{x}))+\boldsymbol{w})-V(\boldsymbol{f}^*(\boldsymbol{x},\boldsymbol{\pi}_s(\boldsymbol{x}))+\boldsymbol{w})] \\
&\quad\leq \mathbb{E}_{\boldsymbol{w}}[\kappa(\|\boldsymbol{f}^*(\boldsymbol{x},\boldsymbol{\pi}(\boldsymbol{x}))+\boldsymbol{w}-(\boldsymbol{f}^*(\boldsymbol{x},\boldsymbol{\pi}_s(\boldsymbol{x}))+\boldsymbol{w})\|)] && \text{(Uniform continuity of } V\text{)} \\
&\quad= \kappa(\|\boldsymbol{f}^*(\boldsymbol{x},\boldsymbol{\pi}(\boldsymbol{x}))-\boldsymbol{f}^*(\boldsymbol{x},\boldsymbol{\pi}_s(\boldsymbol{x}))\|) \\
&\quad\leq \kappa(\kappa_{\boldsymbol{f}^*}(\|\boldsymbol{\pi}(\boldsymbol{x})-\boldsymbol{\pi}_s(\boldsymbol{x})\|)) && \text{(Uniform continuity of } \boldsymbol{f}^*\text{)} \\
&\quad\leq \kappa(\kappa_{\boldsymbol{f}^*}(2u_{\max})). && \text{(Bounded inputs)}
\end{aligned}
$$

Therefore,

$$
\begin{aligned}
\mathbb{E}_{\boldsymbol{x}'|\boldsymbol{\pi},\boldsymbol{x}}[V(\boldsymbol{x}')] &= \mathbb{E}_{\boldsymbol{w}}[V(\boldsymbol{f}^*(\boldsymbol{x},\boldsymbol{\pi}(\boldsymbol{x}))+\boldsymbol{w})] \\
&\leq \mathbb{E}_{\boldsymbol{w}}[V(\boldsymbol{f}^*(\boldsymbol{x},\boldsymbol{\pi}_s(\boldsymbol{x}))+\boldsymbol{w})]+\kappa(\kappa_{\boldsymbol{f}^*}(2u_{\max})) \\
&= \mathbb{E}_{\boldsymbol{x}'|\boldsymbol{\pi}_s,\boldsymbol{x}}[V(\boldsymbol{x}')]+\kappa(\kappa_{\boldsymbol{f}^*}(2u_{\max})) \\
&\leq \gamma V(\boldsymbol{x})+K+\kappa(\kappa_{\boldsymbol{f}^*}(2u_{\max})) \\
&= \gamma V(\boldsymbol{x})+\tilde{K} && (\tilde{K}=K+\kappa(\kappa_{\boldsymbol{f}^*}(2u_{\max})))
\end{aligned}
$$

Hence, $V$ satisfies the drift condition for $\boldsymbol{\pi}$. Furthermore, since $V$ also satisfies positive definiteness by assumption, the bounded energy condition holds for all $\boldsymbol{\pi}\in\Pi$. □

### A.1 Proof of Theorem 2.6

For proving Theorem 2.6, we invoke the results from (Hairer & Mattingly, 2011, Theorem 1.2 – 1.3). For this we require that the Markov chain induced by a policy $\boldsymbol{\pi}$ satisfies the drift condition. In our setting, this corresponds to Assumption 2.4. Next, we show that the chain satisfies the following minorisation condition.

**Lemma A.1** (Minorisation condition)**.** *Consider the system in Equation (1) and let Assumption 2.1 – 2.4 hold. Let $P^{\boldsymbol{\pi}}$ denote the transition kernel for the policy $\boldsymbol{\pi}\in\Pi$, i.e., $P^{\boldsymbol{\pi}}(\boldsymbol{x},\mathcal{A})=\mathbb{P}(\boldsymbol{x}_+\in\mathcal{A}|\boldsymbol{x},\boldsymbol{\pi}(\boldsymbol{x}))$ . Then, for all $\boldsymbol{\pi}\in\Pi$, exists a constant $\alpha\in(0,1)$ and a probability measure $\zeta(\cdot)$ s.t.,*

$$
\inf_{\boldsymbol{x}\in\mathcal{C}} P^{\boldsymbol{\pi}}(\boldsymbol{x},\cdot)\geq\alpha\zeta(\cdot) \tag{9}
$$

*with $\mathcal{C}\stackrel{\text{def}}{=}\{\boldsymbol{x}\in\mathcal{X};V^{\boldsymbol{\pi}}(\boldsymbol{x})\leq R\}$ for some $R>{}^{2K}/_{1-\gamma}$*

*Proof.* We prove it in 3 steps. First, we show that $\mathcal{C}$ is contained in a compact domain. From the Assumption 2.4 we pick the function $\xi\in\mathcal{K}_\infty$. Since $C_l\xi(0)=0,\lim_{s\to\infty}\xi(s)=+\infty$ and $C_l\xi$ is continuous, there exists $M$ such that $C_l\xi(M)=R$. Then for $\|\boldsymbol{x}\|>M$ we have:

$$
V^{\boldsymbol{\pi}}(\boldsymbol{x})\geq C_l\xi(\|\boldsymbol{x}\|)>\xi(M)=R.
$$

Therefore we have: $\mathcal{C}\subseteq\mathcal{B}(\mathbf{0},M)\stackrel{\text{def}}{=}\{\boldsymbol{x}\mid\|\boldsymbol{x}-\mathbf{0}\|\leq M\}$. In the second step we show that $\boldsymbol{f}(\mathcal{C},\boldsymbol{\pi}(\mathcal{C}))$ is bounded, in particular we show that there exists $B>0$ such that: $\boldsymbol{f}(\mathcal{C},\boldsymbol{\pi}(\mathcal{C}))\subseteq\mathcal{B}(\mathbf{0},B)$. This is true since continuous image of compact set is compact and the observation:

$$
\mathcal{C}\subseteq\mathcal{B}(\mathbf{0},M)\implies\boldsymbol{f}(\mathcal{C},\boldsymbol{\pi}(\mathcal{C}))\subseteq\boldsymbol{f}(\mathcal{B}(\mathbf{0},M),\boldsymbol{\pi}(\mathcal{B}(\mathbf{0},M))).
$$

Since $\boldsymbol{f}(\mathcal{B}(\mathbf{0},M),\boldsymbol{\pi}(\mathcal{B}(\mathbf{0},M)))$ is compact there exists $B$ such that $\boldsymbol{f}(\mathcal{C},\boldsymbol{\pi}(\mathcal{C}))\subseteq\mathcal{B}(\mathbf{0},B)$. In the last step we prove that $\alpha\stackrel{\text{def}}{=}2^{-d_{\boldsymbol{x}}}e^{-B^2/\sigma^2}$ and $\zeta$ with law of $\mathcal{N}\left(0,\frac{\sigma^2}{2}\right)$ satisfy condition of Lemma A.1. It is enough to show that $\forall\boldsymbol{\mu}\in\mathcal{B}(\mathbf{0},B),\forall\boldsymbol{x}\in\mathbb{R}^{d_{\boldsymbol{x}}}$ we have:

$$
\alpha\frac{1}{(2\pi)^{\frac{d_{\boldsymbol{x}}}{2}}\left(\frac{\sigma^2}{2}\right)^{\frac{d_{\boldsymbol{x}}}{2}}}e^{-\frac{\|\boldsymbol{x}\|^2}{\sigma^2}}\leq\frac{1}{(2\pi)^{\frac{d_{\boldsymbol{x}}}{2}}(\sigma^2)^{\frac{d_{\boldsymbol{x}}}{2}}}e^{-\frac{\|\boldsymbol{x}-\boldsymbol{\mu}\|^2}{2\sigma^2}}
$$

which can be proven with simple algebraic manipulations. □

Through the minorisation condition and Assumption 2.4, we can prove the ergodicity of the closed-loop system for a given policy $\boldsymbol{\pi} \in \Pi$.

**Theorem A.2** (Ergodicity of closed-loop system)**.** *Let Assumption 2.1 – 2.4, consider any probability measures $\zeta_1$, $\zeta_2$, and $\theta > 0$, define $P^{\boldsymbol{\pi}}\zeta$, $\|\varphi\|_{1+\theta V^{\boldsymbol{\pi}}}$, $\rho_\theta^{\boldsymbol{\pi}}$ as*

$$\begin{aligned}
\left(P^{\boldsymbol{\pi}}\zeta\right)(\mathcal{A}) &= \int_{\mathcal{X}} P^{\boldsymbol{\pi}}(\boldsymbol{x}, \mathcal{A})\zeta(d\boldsymbol{x}) \\
\|\varphi\|_{1+\theta V^{\boldsymbol{\pi}}} &= \sup_{\boldsymbol{x}\in\mathcal{X}} \frac{|\varphi(\boldsymbol{x})|}{1+\theta V^{\boldsymbol{\pi}}(\boldsymbol{x})} \\
\rho_\theta^{\boldsymbol{\pi}}(\zeta_1,\zeta_2) &= \sup_{\varphi:\|\varphi\|_{1+\theta V^{\boldsymbol{\pi}}}\le 1} \int_{\mathcal{X}} \varphi(\boldsymbol{x})(\zeta_1-\zeta_2)(d\boldsymbol{x}) = \int_{\mathcal{X}} (1+\theta V^{\boldsymbol{\pi}}(\boldsymbol{x}))|\zeta_1-\zeta_2|(d\boldsymbol{x}).
\end{aligned}$$

*We have for all $\boldsymbol{\pi} \in \Pi$, that $P^{\boldsymbol{\pi}}$ admits a unique invariant measure $\bar{P}^{\boldsymbol{\pi}}$. Furthermore, there exist constants $C_1 > 0$, $\theta > 0$, $\lambda \in (0,1)$ such that*

$$\rho_\theta^{\boldsymbol{\pi}}(P^{\boldsymbol{\pi}}\zeta_1, P^{\boldsymbol{\pi}}\zeta_2) \le \lambda\rho_\theta^{\boldsymbol{\pi}}(\zeta_1,\zeta_2) \tag{1}$$

$$\left\|\mathbb{E}_{\boldsymbol{x}\sim(P^{\boldsymbol{\pi}})^t}\left[\varphi(\boldsymbol{x})\right] - \mathbb{E}_{\boldsymbol{x}\sim\bar{P}^{\boldsymbol{\pi}}}\left[\varphi(\boldsymbol{x})\right]\right\|_{1+V^{\boldsymbol{\pi}}} \le C_1\lambda^t \left\|\varphi - \mathbb{E}_{\boldsymbol{x}\sim\bar{P}^{\boldsymbol{\pi}}}\left[\varphi(\boldsymbol{x})\right]\right\|_{1+V^{\boldsymbol{\pi}}}. \tag{2}$$

*holds for every measurable function $\varphi : \mathcal{X} \to \mathcal{R}$ with $\|\varphi\|_{1+V^{\boldsymbol{\pi}}} < \infty$. Here $(P^{\boldsymbol{\pi}})^t$ denotes the $t$-step transition kernel under the policy $\boldsymbol{\pi}$.*
*Moreover, $\theta = \alpha_0/K$, and*

$$\lambda = \max\left\{1-(\alpha-\alpha_0), \frac{2+R/K\alpha_0\gamma_0}{2+R/K\alpha_0}\right\} \tag{10}$$

*for any $\alpha_0 \in (0,\alpha)$ and $\gamma_0 \in (\gamma + 2K/R, 1)$.*

*Proof.* From Assumption 2.4, we have a value function for each policy that satisfies the drift condition. Furthermore, in Lemma A.1 we show that our system also satisfies the minorisation condition for all policies. Under these conditions, we can use the results from Hairer & Mattingly (2011, Theorem 1.2. – 1.3.). □

Note that $\|\cdot\|_{1+\theta V^{\boldsymbol{\pi}}}$ represents a family of equivalent norms for any $\theta > 0$. Now we prove Theorem 2.6.

*Proof of Theorem 2.6.* From Theorem A.2, we have

$$\rho_\theta^{\boldsymbol{\pi}}((P^{\boldsymbol{\pi}})^{t+1},(P^{\boldsymbol{\pi}})^t) = \rho_\theta^{\boldsymbol{\pi}}(P^{\boldsymbol{\pi}}(P^{\boldsymbol{\pi}})^t, P^{\boldsymbol{\pi}}(P^{\boldsymbol{\pi}})^{t-1}) \le \lambda^t\rho_\theta^{\boldsymbol{\pi}}(P^{\boldsymbol{\pi}}\delta_{\boldsymbol{x}_0},\delta_{\boldsymbol{x}_0}),$$

where $\delta_{\boldsymbol{x}_0}$ is the dirac measure. Therefore, $(P^{\boldsymbol{\pi}})^t$ is a Cauchy sequence. Furthermore, $\rho_\theta^{\boldsymbol{\pi}}$ is complete for the set of probability measures integrating $V$, thus $\rho_\theta^{\boldsymbol{\pi}}((P^{\boldsymbol{\pi}})^t,\bar{P}^{\boldsymbol{\pi}}) \to 0$ for $t\to\infty$ (c.f., Hairer & Mattingly (2011) for more details). In particular, we have for $\varphi$ such that $\|\varphi\|_{1+\theta V^{\boldsymbol{\pi}}} \le 1$,

$$\lim_{t\to\infty}\int_{\mathcal{X}}\varphi(\boldsymbol{x})(P^{\boldsymbol{\pi}})^t(d\boldsymbol{x}) = \int_{\mathcal{X}}\varphi(\boldsymbol{x})\bar{P}^{\boldsymbol{\pi}}(d\boldsymbol{x}).$$

Note that since all $\|\cdot\|_{1+\theta V^{\boldsymbol{\pi}}}$ norms are equivalent for $\theta > 0$, if $\|c\|_{1+V^{\boldsymbol{\pi}}} \le C$ (Assumption 2.4), then $\|c\|_{1+\theta V^{\boldsymbol{\pi}}} \le C'$ for some $C' \in (0,\infty)$. Furthermore, note that $c(\cdot) \ge 0$. Therefore,

$$\begin{aligned}
\int_{\mathcal{X}} c(\boldsymbol{x})\bar{P}^{\boldsymbol{\pi}}(d\boldsymbol{x}) &= \lim_{t\to\infty}\int_{\mathcal{X}} c(\boldsymbol{x})(P^{\boldsymbol{\pi}})^t(d\boldsymbol{x}) \\
&\le C\lim_{t\to\infty}\int_{\mathcal{X}}(1+V^{\boldsymbol{\pi}}(\boldsymbol{x}))(P^{\boldsymbol{\pi}})^t(d\boldsymbol{x}) \\
&= C + C\lim_{t\to\infty}\mathbb{E}_{\boldsymbol{x}\sim(P^{\boldsymbol{\pi}})^t}[V^{\boldsymbol{\pi}}(\boldsymbol{x})] \\
&= C + C\lim_{t\to\infty}\mathbb{E}_{\boldsymbol{x}\sim(P^{\boldsymbol{\pi}})^{t-1}}[\mathbb{E}_{\boldsymbol{x}'\sim(P^{\boldsymbol{\pi}})}[V^{\boldsymbol{\pi}}(\boldsymbol{x}')|\boldsymbol{x}]] \\
&\le C + C\left(\lim_{t\to\infty}\gamma\mathbb{E}_{\boldsymbol{x}\sim(P^{\boldsymbol{\pi}})^{t-1}}[V^{\boldsymbol{\pi}}(\boldsymbol{x})] + K\right) && \text{(Assumption 2.4)}
\end{aligned}$$

$$
\begin{aligned}
&\leq C + C \lim_{t\to\infty} \gamma^t V^{\boldsymbol{\pi}}(\boldsymbol{x}_0) + K\frac{1-\gamma^t}{1-\gamma} \\
&= C\left(1 + K\frac{1}{1-\gamma}\right)
\end{aligned}
$$

In summary, we have $\mathbb{E}_{\boldsymbol{x}\sim\bar{P}^{\boldsymbol{\pi}}}\left[c(\boldsymbol{x})\right] \leq C\left(1 + K\frac{1}{1-\gamma}\right)$

Consider any $t > 0$, and note that from Theorem A.2 we have

$$
\begin{aligned}
\left\|\mathbb{E}_{\boldsymbol{x}\sim(P^{\boldsymbol{\pi}})^t}\left[c(\boldsymbol{x})\right] - \mathbb{E}_{\boldsymbol{x}\sim\bar{P}^{\boldsymbol{\pi}}}\left[c(\boldsymbol{x})\right]\right\|_{1+V^{\boldsymbol{\pi}}} &= \sup_{\boldsymbol{x}_0\in\mathcal{X}} \frac{\left|\mathbb{E}_{\boldsymbol{x}\sim(P^{\boldsymbol{\pi}})^t}\left[c(\boldsymbol{x})\right] - \mathbb{E}_{\boldsymbol{x}\sim\bar{P}^{\boldsymbol{\pi}}}\left[c(\boldsymbol{x})\right]\right|}{1+V^{\boldsymbol{\pi}}(\boldsymbol{x}_0)} \\
&\leq C_1\lambda^t \left\|c - \mathbb{E}_{\boldsymbol{x}\sim\bar{P}^{\boldsymbol{\pi}}}\left[c(\boldsymbol{x})\right]\right\|_{1+V^{\boldsymbol{\pi}}} \qquad \text{(Theorem A.2)} \\
&\leq C_1\lambda^t \left\|c\right\|_{1+V^{\boldsymbol{\pi}}} + C_1\lambda^t\mathbb{E}_{\boldsymbol{x}\sim\bar{P}^{\boldsymbol{\pi}}}\left[c(\boldsymbol{x})\right] \\
&= C_2\lambda^t,
\end{aligned}
$$

where $C_2 = C_1(\|c\|_{1+V^{\boldsymbol{\pi}}} + CK\frac{1}{1-\gamma})$.

Moreover, since the inequality holds for all $\boldsymbol{x}_0$, we have

$$
\frac{\left|\mathbb{E}_{\boldsymbol{x}\sim(P^{\boldsymbol{\pi}})^t}\left[c(\boldsymbol{x})\right] - \mathbb{E}_{\boldsymbol{x}\sim\bar{P}^{\boldsymbol{\pi}}}\left[c(\boldsymbol{x})\right]\right|}{1+V^{\boldsymbol{\pi}}(\boldsymbol{x}_0)} \leq C_2\lambda^t.
$$

In summary,

$$
\left|\mathbb{E}_{\boldsymbol{x}\sim(P^{\boldsymbol{\pi}})^t}\left[c(\boldsymbol{x})\right] - \mathbb{E}_{\boldsymbol{x}\sim\bar{P}^{\boldsymbol{\pi}}}\left[c(\boldsymbol{x})\right]\right| \leq C_2(1+V^{\boldsymbol{\pi}}(\boldsymbol{x}_0))\lambda^t.
$$

Consider any $T \geq 0$, and define with $\bar{c} = \mathbb{E}_{\boldsymbol{x}\sim\bar{P}^{\boldsymbol{\pi}}}\left[c(\boldsymbol{x}, \boldsymbol{\pi}(x))\right]$.

$$
\begin{aligned}
\mathbb{E}_{\boldsymbol{\pi}}\left[\sum_{t=0}^{T-1} c(\boldsymbol{x}_t, \boldsymbol{u}_t) - \bar{c}\right] &= \sum_{t=0}^{T-1} \mathbb{E}_{(P^{\boldsymbol{\pi}})^t}\left[c(\boldsymbol{x}_t, \boldsymbol{u}_t)\right] - \bar{c} \\
&\leq \sum_{t=0}^{T-1} \left|\mathbb{E}_{(P^{\boldsymbol{\pi}})^t}\left[c(\boldsymbol{x}_t, \boldsymbol{u}_t)\right] - \bar{c}\right| \\
&\leq C_2(1+V^{\boldsymbol{\pi}}(\boldsymbol{x}_0)) \sum_{t=0}^{T-1} \lambda^t \\
&= C_2(1+V^{\boldsymbol{\pi}}(\boldsymbol{x}_0)) \frac{1-\lambda^T}{1-\lambda}
\end{aligned}
$$

Hence, we have

$$
\lim_{T\to\infty} \left|\mathbb{E}_{\boldsymbol{\pi}}\left[\sum_{t=0}^{T-1} c(\boldsymbol{x}_t, \boldsymbol{u}_t) - \bar{c}\right]\right| \leq C_2(1+V^{\boldsymbol{\pi}}(\boldsymbol{x}_0))\frac{1}{1-\lambda},
$$

and for any $\boldsymbol{x}_0$ in a compact subset of $\mathcal{X}$

$$
\lim_{T\to\infty} \frac{1}{T}\mathbb{E}_{\boldsymbol{\pi}}\left[\sum_{t=0}^{T-1} c(\boldsymbol{x}_t, \boldsymbol{u}_t) - \bar{c}\right] = 0.
$$

Moreover,

$$
|B(\boldsymbol{\pi}, \boldsymbol{x}_0)| \leq C_2(1+V^{\boldsymbol{\pi}}(\boldsymbol{x}_0))\frac{1}{1-\lambda}.
$$

□

Another interesting, inequality that follows from the proof above is the difference in bias inequality.

$$
\left|\mathbb{E}_{\boldsymbol{x}_0\sim\zeta_1}[B(\boldsymbol{\pi}, \boldsymbol{x}_0)] - \mathbb{E}_{\boldsymbol{x}_0\sim\zeta_2}[B(\boldsymbol{\pi}, \boldsymbol{x}_0)]\right| \leq \frac{C_3}{1-\lambda}\int_{\mathcal{X}} (1+V^{\boldsymbol{\pi}}(\boldsymbol{x}))\left|\zeta_1 - \zeta_2\right|(d\boldsymbol{x})
$$

for all probability measures $\zeta_1, \zeta_2$. To show this holds, define $C' = \max_{\boldsymbol{\pi} \in \Pi} \|c(\boldsymbol{x}, \boldsymbol{\pi}(\boldsymbol{x}))\|_{1+\theta V^{\boldsymbol{\pi}}}$. Furthermore, note that $C' < \infty$ from Assumption 2.4 and $\|{}^{c(\boldsymbol{x},\boldsymbol{\pi}(\boldsymbol{x}))}/_{C'}\|_{1+\theta V^{\boldsymbol{\pi}}} \leq 1$.

$$\begin{aligned}
&\left|\mathbb{E}_{\boldsymbol{x}\sim(P^{\boldsymbol{\pi}})^t\zeta_1} c(\boldsymbol{x}, \boldsymbol{\pi}(\boldsymbol{x})) - \mathbb{E}_{\boldsymbol{x}\sim(P^{\boldsymbol{\pi}})^t\zeta_2} c(\boldsymbol{x}, \boldsymbol{\pi}(\boldsymbol{x}))\right| = \left|\int_{\mathcal{X}} c(\boldsymbol{x}, \boldsymbol{\pi}(\boldsymbol{x}))((P^{\boldsymbol{\pi}})^t\zeta_1 - (P^{\boldsymbol{\pi}})^t\zeta_2)(d\boldsymbol{x})\right| \\
&= C' \left|\int_{\mathcal{X}} \frac{1}{C'} c(\boldsymbol{x}, \boldsymbol{\pi}(\boldsymbol{x}))((P^{\boldsymbol{\pi}})^t\zeta_1 - (P^{\boldsymbol{\pi}})^t\zeta_2)(d\boldsymbol{x})\right| \\
&\leq C' \sup_{\varphi: \|\varphi\|_{1+\theta V^{\boldsymbol{\pi}}} \leq 1} \int_{\mathcal{X}} \varphi(\boldsymbol{x})((P^{\boldsymbol{\pi}})^t\zeta_1 - (P^{\boldsymbol{\pi}})^t\zeta_2)(d\boldsymbol{x}) = C' \rho_\theta^{\boldsymbol{\pi}}((P^{\boldsymbol{\pi}})^t\zeta_1, (P^{\boldsymbol{\pi}})^t\zeta_2) \\
&\leq C' \lambda \rho_\theta^{\boldsymbol{\pi}}((P^{\boldsymbol{\pi}})^{t-1}\zeta_1, (P^{\boldsymbol{\pi}})^{t-1}\zeta_2) \qquad \text{(Theorem A.2)} \\
&\leq C' \lambda^t \rho_\theta^{\boldsymbol{\pi}}(\zeta_1, \zeta_2).
\end{aligned}$$

Also, note that there exists $C_\theta \in (0, \infty)$ such that $C_\theta \|\varphi\|_{1+\theta V^{\boldsymbol{\pi}}} \geq \|\varphi\|_{1+V^{\boldsymbol{\pi}}}$ due to the equivalence of the two norms.

$$\begin{aligned}
\rho_\theta^{\boldsymbol{\pi}}(\zeta_1, \zeta_2) &= \sup_{\varphi: \|\varphi\|_{1+\theta V^{\boldsymbol{\pi}}} \leq 1} \int_{\mathcal{X}} \varphi(\boldsymbol{x})(\zeta_1 - \zeta_2)(d\boldsymbol{x}) \\
&\leq \sup_{\varphi: \|\varphi\|_{1+V^{\boldsymbol{\pi}}} \leq C_\theta} \int_{\mathcal{X}} \varphi(\boldsymbol{x})(\zeta_1 - \zeta_2)(d\boldsymbol{x}) \\
&= C_\theta \sup_{\varphi: \|\varphi\|_{1+V^{\boldsymbol{\pi}}} \leq 1} \int_{\mathcal{X}} \varphi(\boldsymbol{x})(\zeta_1 - \zeta_2)(d\boldsymbol{x}) \\
&= C_\theta \rho_1^{\boldsymbol{\pi}}(\zeta_1, \zeta_2)
\end{aligned}$$

Therefore, for the bias we have

$$\begin{aligned}
&|\mathbb{E}_{\boldsymbol{x}_0\sim\zeta_1}[B(\boldsymbol{\pi}, \boldsymbol{x}_0)] - \mathbb{E}_{\boldsymbol{x}_0\sim\zeta_2}[B(\boldsymbol{\pi}, \boldsymbol{x}_0)]| \\
&\leq \lim_{T\to\infty} \sum_{t=0}^{T-1} \left|\mathbb{E}_{\boldsymbol{x}\sim(P^{\boldsymbol{\pi}})^t\zeta_1} c(\boldsymbol{x}, \boldsymbol{\pi}(\boldsymbol{x})) - \mathbb{E}_{\boldsymbol{x}\sim(P^{\boldsymbol{\pi}})^t\zeta_2} c(\boldsymbol{x}, \boldsymbol{\pi}(\boldsymbol{x}))\right| \\
&\leq C' \rho_\theta^{\boldsymbol{\pi}}(\zeta_1, \zeta_2) \lim_{T\to\infty} \sum_{t=0}^{T-1} \lambda^t = \frac{C'}{1-\lambda} \rho_\theta^{\boldsymbol{\pi}}(\zeta_1, \zeta_2) \\
&\leq \frac{C' C_\theta}{1-\lambda} \rho_1^{\boldsymbol{\pi}}(\zeta_1, \zeta_2) = \frac{C' C_\theta}{1-\lambda} \int_{\mathcal{X}} (1 + V^{\boldsymbol{\pi}}(\boldsymbol{x})) \left|\zeta_1 - \zeta_2\right| (d\boldsymbol{x})
\end{aligned}$$

Set $C_3 = C' C_\theta$.

### A.2 Proof of bounded average cost for the optimistic system

In this section, we show that the results from Theorem 2.6 also transfer over to the optimistic dynamics.

**Theorem A.3** (Existence of Average Cost Solution for the Optimistic System)**.** *Let Assumption 2.1 – 2.8 hold. Consider any $n > 0$ and let $\boldsymbol{\pi}_n, \boldsymbol{f}_n$ denote the solution to Equation (6), $P^{\boldsymbol{\pi}, \boldsymbol{f}_n}$ its transition kernel. Then $P^{\boldsymbol{\pi}, \boldsymbol{f}_n}$ admits a unique invariant measure $\bar{P}^{\boldsymbol{\pi}_n, \boldsymbol{f}_n}$ and there exists $C_2, C_3 \in (0, \infty)$, $\hat{\lambda} \in (0, 1)$ such that*

Average Cost*;*

$$A(\boldsymbol{\pi}_n, \boldsymbol{f}_n) = \lim_{T\to\infty} \frac{1}{T} \mathbb{E}_{\boldsymbol{\pi}_n, \boldsymbol{f}_n}\left[\sum_{t=0}^{T-1} c(\boldsymbol{x}_t, \boldsymbol{u}_t)\right] = \mathbb{E}_{\boldsymbol{x}\sim\bar{P}^{\boldsymbol{\pi}_n, \boldsymbol{f}_n}}\left[c(\boldsymbol{x}, \boldsymbol{\pi}_n(x))\right]$$

Bias Cost*;*

$$|B(\boldsymbol{\pi}_n, \boldsymbol{f}_n, \boldsymbol{x}_0)| = \left|\lim_{T\to\infty} \mathbb{E}_{\boldsymbol{\pi}_n, \boldsymbol{f}_n}\left[\sum_{t=0}^{T-1} c(\boldsymbol{x}_t, \boldsymbol{u}_t) - A(\boldsymbol{\pi}_n, \boldsymbol{f}_n)\right]\right| \leq C_2(1 + V^{\boldsymbol{\pi}_n}(\boldsymbol{x}_0)) \frac{1}{1-\hat{\lambda}}$$

*for all $\boldsymbol{x}_0 \in \mathcal{X}$.*

Difference in Bias*;*

$$|\mathbb{E}_{\boldsymbol{x}_0\sim\zeta_1}[B(\boldsymbol{\pi}_n,\boldsymbol{f}_n,\boldsymbol{x}_0)]-\mathbb{E}_{\boldsymbol{x}_0\sim\zeta_2}[B(\boldsymbol{\pi}_n,\boldsymbol{f}_n,\boldsymbol{x}_0)]|\leq\frac{C_3}{1-\hat{\lambda}}\int_{\mathcal{X}}(1+V^{\boldsymbol{\pi}}(\boldsymbol{x}))\,|\zeta_1-\zeta_2|\,(d\boldsymbol{x})$$

*for all probability measures* $\zeta_1,\zeta_2$*.*

Theorem A.3 shows that the optimistic dynamics $\boldsymbol{f}_n$ retain the boundedness property from the true dynamics $\boldsymbol{f}^*$ and give a well-defined solution w.r.t. average cost and the bias cost. To prove Theorem A.3 we show that the optimistic system also satisfies the drift and minorisation condition. Then we can invoke the result from Hairer & Mattingly (2011) similar to the proof of Theorem 2.6.

**Lemma A.4** (Stability of optimistic system)**.** *Let Assumption 2.1 – 2.8 hold, then we have with probability at least* $1-\delta$ *for all* $n\geq 0$*,* $\boldsymbol{\pi}\in\Pi$*,* $\boldsymbol{f}\in\mathcal{M}_n\cap\mathcal{M}_0$*, that there exists a constant* $\widehat{K}>0$ *such that*

$$\mathbb{E}_{\boldsymbol{x}_+|\boldsymbol{x},\boldsymbol{f},\boldsymbol{\pi}}[V^{\boldsymbol{\pi}}(\boldsymbol{x}_+)]\leq\gamma V^{\boldsymbol{\pi}}(\boldsymbol{x})+\widehat{K},$$

*where* $\boldsymbol{x}_+=\boldsymbol{f}(\boldsymbol{x},\boldsymbol{\pi}(\boldsymbol{x})+\boldsymbol{w}$*.*

*Proof.* Note, that $V^{\boldsymbol{\pi}}$ is uniformly continuous w.r.t. $\kappa$

$$|V^{\boldsymbol{\pi}}(\boldsymbol{x})-V^{\boldsymbol{\pi}}(\boldsymbol{x}')|\leq\kappa(\|\boldsymbol{x}-\boldsymbol{x}'\|).$$

Furthermore, since $\boldsymbol{f}\in\mathcal{M}_n\cap\mathcal{M}_0$ and therefore $\boldsymbol{f}\in\mathcal{M}_0$, we have that there exists some $\boldsymbol{\eta}\in[-1,1]^{d_x}$ such that

$$\boldsymbol{f}(\boldsymbol{x},\boldsymbol{\pi}(\boldsymbol{x}))=\boldsymbol{\mu}_0(\boldsymbol{x}.\boldsymbol{\pi}(\boldsymbol{x}))+\beta_0\boldsymbol{\sigma}_0(\boldsymbol{x},\boldsymbol{\pi}(\boldsymbol{x}))\boldsymbol{\eta}(\boldsymbol{x}).$$

$$\begin{aligned}
&\mathbb{E}_{\boldsymbol{w}}[V^{\boldsymbol{\pi}}(\boldsymbol{\mu}_0(\boldsymbol{x}.\boldsymbol{\pi}(\boldsymbol{x}))+\beta_0\boldsymbol{\sigma}_0(\boldsymbol{x},\boldsymbol{\pi}(\boldsymbol{x}))\boldsymbol{\eta}(\boldsymbol{x})+\boldsymbol{w})]-\mathbb{E}_{\boldsymbol{w}}[V^{\boldsymbol{\pi}}(\boldsymbol{f}^*(\boldsymbol{x}.\boldsymbol{\pi}(\boldsymbol{x}))+\boldsymbol{w})]\\
&\quad\leq\kappa\left(\|\boldsymbol{\mu}_0(\boldsymbol{x}.\boldsymbol{\pi}(\boldsymbol{x}))+\beta_0\boldsymbol{\sigma}_0(\boldsymbol{x},\boldsymbol{\pi}(\boldsymbol{x}))\boldsymbol{\eta}(\boldsymbol{x})-\boldsymbol{f}^*(\boldsymbol{x}.\boldsymbol{\pi}(\boldsymbol{x}))\|\right)\\
&\quad\leq\kappa\left(\|\boldsymbol{\mu}_0(\boldsymbol{x}.\boldsymbol{\pi}(\boldsymbol{x}))-\boldsymbol{f}^*(\boldsymbol{x}.\boldsymbol{\pi}(\boldsymbol{x}))\|+\|\beta_0\boldsymbol{\sigma}_0(\boldsymbol{x},\boldsymbol{\pi}(\boldsymbol{x}))\boldsymbol{\eta}(\boldsymbol{x})\|\right)\\
&\quad\leq\kappa\left(\left(1+\sqrt{d_x}\right)\beta_0\sqrt{d_x}\sigma_{\max}\right). && \text{(Assumption 2.8)}
\end{aligned}$$

Therefore,

$$\begin{aligned}
\mathbb{E}_{\boldsymbol{x}_+|\boldsymbol{x},\boldsymbol{f},\boldsymbol{\pi}}[V^{\boldsymbol{\pi}}(\boldsymbol{x}_+)]&\leq\mathbb{E}_{\boldsymbol{x}_+|\boldsymbol{x},\boldsymbol{f}^*,\boldsymbol{\pi}}[V^{\boldsymbol{\pi}}(\boldsymbol{x}^*_+)]+\kappa\left(\left(1+\sqrt{d_x}\right)\beta_0\sqrt{d_x}\sigma_{\max}\right)\\
&=\mathbb{E}_{\boldsymbol{x}_+|\boldsymbol{x},\boldsymbol{f}^*,\boldsymbol{\pi}}[V^{\boldsymbol{\pi}}(\boldsymbol{x}^*_+)]+\kappa\left(\left(1+\sqrt{d_x}\right)\beta_0\sqrt{d_x}\sigma_{\max}\right)\\
&\leq\gamma V^{\boldsymbol{\pi}}(\boldsymbol{x})+K+\kappa\left(\left(1+\sqrt{d_x}\right)\beta_0\sqrt{d_x}\sigma_{\max}\right), && \text{(Assumption 2.4)}
\end{aligned}$$

where we denoted $\boldsymbol{x}^*_+=\boldsymbol{f}^*(\boldsymbol{x},\boldsymbol{\pi}(\boldsymbol{x})+\boldsymbol{w}$. Define $\widehat{K}=K+\kappa\left(\left(1+\sqrt{d_x}\right)\beta_0\sqrt{d_x}\sigma_{\max}\right)$. □

**Lemma A.5** (Minorisation condition optimistic ystem)**.** *Consider the system*

$$\boldsymbol{x}_+=\boldsymbol{f}(\boldsymbol{x}.\boldsymbol{\pi}(\boldsymbol{x}))+\boldsymbol{w}$$

*for any* $n\geq 0$*,* $\boldsymbol{\pi}\in\Pi$ *and* $\boldsymbol{f}\in\mathcal{M}_n\cap\mathcal{M}_0$*. Let Assumption 2.1 – 2.8 hold. Let* $P^{\boldsymbol{\pi},\boldsymbol{f}}$ *denote the transition kernel for the policy* $\boldsymbol{\pi}\in\Pi$ *i.e.,* $P^{\boldsymbol{\pi},\boldsymbol{f}}(\boldsymbol{x},\mathcal{A})=\mathbb{P}(\boldsymbol{x}_+\in\mathcal{A}|\boldsymbol{x},\boldsymbol{\pi}(\boldsymbol{x}),\boldsymbol{f})$*. Then, there exists a constant* $\hat{\alpha}\in(0,1)$ *and a probability measure* $\hat{\zeta}(\cdot)$ *independent of* $n$ *s.t.,*

$$\inf_{\boldsymbol{x}\in\mathcal{C}}P^{\boldsymbol{\pi},\boldsymbol{f}}(\boldsymbol{x},\cdot)\geq\hat{\alpha}\hat{\zeta}(\cdot)\tag{11}$$

*with* $\mathcal{C}\stackrel{\text{def}}{=}\{\boldsymbol{x}\in\mathcal{X};V^{\boldsymbol{\pi}}(\boldsymbol{x})<\hat{R}\}$ *for some* $\hat{R}>{}^{2\widehat{K}}/_{1-\gamma}$

*Proof.* First, we show that $\mathcal{C}$ is contained in a compact domain. From the Assumption 2.4 we pick the function $\xi\in\mathcal{K}_\infty$. Since $C_l\xi(0)=0,\lim_{s\to\infty}\xi(s)=+\infty$ and $C_l\xi$ is continuous, there exists $M$ such that $C_l\xi(M)=\hat{R}$. Then for $\|\boldsymbol{x}\|>M$ we have:

$$V^{\boldsymbol{\pi}}(\boldsymbol{x})\geq C_l\xi(\|\boldsymbol{x}\|)>\xi(M)=\hat{R}.$$

Therefore we have: $\mathcal{C} \subseteq \mathcal{B}(\mathbf{0}, M) \stackrel{\text{def}}{=} \{\boldsymbol{x} \mid \|\boldsymbol{x} - \mathbf{0}\| \le M\}$. Since for any $\boldsymbol{x} \in \mathcal{C}$ we have $\|\boldsymbol{f}(\boldsymbol{x}, \boldsymbol{\pi}(\boldsymbol{x}))\| \le \|\boldsymbol{f}^*(\boldsymbol{x}, \boldsymbol{\pi}(\boldsymbol{x}))\| + \beta_0 \sigma_{\max}$. Since $\boldsymbol{f}^*$ is continuous, there exists a $B$ such that $\boldsymbol{f}^*(\mathcal{C}, \boldsymbol{\pi}(\mathcal{C})) \subset \mathcal{B}(\mathbf{0}, B)$. Therefore we have: $\boldsymbol{f}(\mathcal{C}, \boldsymbol{\pi}(\mathcal{C})) \subset \mathcal{B}(\mathbf{0}, B_1)$, where $B_1 = B + \beta_0 \sigma_{\max}$. In the last step we prove that $\alpha \stackrel{\text{def}}{=} 2^{-d_{\boldsymbol{x}}} e^{-B_1^2/\sigma^2}$ and $\zeta$ with law of $\mathcal{N}\left(0, \frac{\sigma^2}{2}\right)$ satisfy condition of Lemma A.1. It is enough to show that $\forall \boldsymbol{\mu} \in \mathcal{B}(\mathbf{0}, B_1), \forall \boldsymbol{x} \in \mathbb{R}^{d_{\boldsymbol{x}}}$ we have:

$$\alpha \frac{1}{(2\pi)^{\frac{d_{\boldsymbol{x}}}{2}} \left(\frac{\sigma^2}{2}\right)^{\frac{d_{\boldsymbol{x}}}{2}}} e^{-\frac{\|\boldsymbol{x}\|^2}{\sigma^2}} \le \frac{1}{(2\pi)^{\frac{d_{\boldsymbol{x}}}{2}} (\sigma^2)^{\frac{d_{\boldsymbol{x}}}{2}}} e^{-\frac{\|\boldsymbol{x} - \boldsymbol{\mu}\|^2}{2\sigma^2}}$$

which can be proven with simple algebraic manipulations. □

*Proof of Theorem A.3.* As for the true system, the drift condition from Lemma A.4 and the minorisation condition from Lemma A.5 are sufficient to show ergodicity of the optimistic system (c.f., Theorem A.2 or Hairer & Mattingly (2011)). The rest of the proof is similar to Theorem 2.6. □

### A.3 Proof of Theorem 3.1

Since NeoRL works in artificial episodes $n \in \{0, N-1\}$ of varying horizons $H_n$. We denote with $\boldsymbol{x}_k^n$ the state visited during episode $n$ at time step $k \le H_n$. Crucial, to our regret analysis is bounding the first and second moment of $V^{\boldsymbol{\pi}_n}(\boldsymbol{x}_k^n)$ for all $n, k$. Given the nature of Assumption 2.4, this requires analyzing geometric series. Thus, we start with the following elementary result of geometric series.

**Corollary A.6.** *Consider the sequence $\{S_n\}_{n \ge 0}$ with $S_n \ge 0$ for all $n$. Let the following hold*

$$S_n \le \rho S_{n-1} + C$$

*for $\rho \in (0, 1)$ and $C > 0$. Then we have*

$$S_n \le \rho^n S_0 + C \frac{1}{1 - \rho}.$$

*Proof.*

$$S_n \le \rho S_{n-1} + C \le \rho^2 S_{n-2} + C(1 + \rho) \le \rho^n S_0 + C \sum_{i=0}^{n} \rho^i \le \rho^n S_0 + C \frac{1}{1 - \rho}.$$

□

**Lemma A.7.** *Let Assumption 2.1 – 2.8 hold and let $H_0$ be the smallest integer such that*

$$H_0 > \frac{\log\left(C_u/C_l\right)}{\log\left(1/\gamma\right)}.$$

*Moreover, define $\nu = \frac{C_u}{C_l} \gamma^{H_0}$. Note, by definition of $H_0$, $\nu < 1$. Then we have for all $k \in \{0, \dots, H_n\}$ and $n > 0$*

Bounded expectation over horizon

$$\mathbb{E}_{\boldsymbol{x}_k^n, \dots, \boldsymbol{x}_1^0 | \boldsymbol{x}_0}[V^{\boldsymbol{\pi}_n}(\boldsymbol{x}_k^n)] \le \gamma^k \mathbb{E}_{\boldsymbol{x}_0^n, \dots, \boldsymbol{x}_1^0 | \boldsymbol{x}_0}[V^{\boldsymbol{\pi}_n}(\boldsymbol{x}_0^n)] + K/(1 - \gamma). \tag{12}$$

Bounded expectation over episodes

$$\mathbb{E}_{\boldsymbol{x}_0^n, \dots, \boldsymbol{x}_1^0 | \boldsymbol{x}_0}[V^{\boldsymbol{\pi}_n}(\boldsymbol{x}_0^n)] \le \nu^n V^{\boldsymbol{\pi}_0}(\boldsymbol{x}_0) + \frac{C_u}{C_l} K/(1 - \gamma) \frac{1}{1 - \nu}. \tag{13}$$

*Moreover, we have*

$$\mathbb{E}_{\boldsymbol{x}_k^n, \dots, \boldsymbol{x}_1^0 | \boldsymbol{x}_0}[V^{\boldsymbol{\pi}_n}(\boldsymbol{x}_k^n)] \le D(\boldsymbol{x}_0, K, \gamma, \nu), \tag{14}$$

*with $D(\boldsymbol{x}_0, K, \gamma, \nu) = V^{\boldsymbol{\pi}_0}(\boldsymbol{x}_0) + K/(1 - \gamma)\left(\frac{C_u}{C_l} \frac{1}{1 - \nu} + 1\right)$*

*Proof.* We start with proving the first claim

$$\begin{aligned}
\mathbb{E}_{\boldsymbol{x}_k^n,\ldots,\boldsymbol{x}_1^0|\boldsymbol{x}_0}[V^{\boldsymbol{\pi}_n}(\boldsymbol{x}_k^n)] &= \mathbb{E}_{\boldsymbol{x}_{k-1}^n,\ldots,\boldsymbol{x}_1^0|\boldsymbol{x}_0}[\mathbb{E}_{\boldsymbol{x}_k^n|\boldsymbol{x}_{k-1}^n}[V^{\boldsymbol{\pi}_n}(\boldsymbol{x}_k^n)]] \\
&\leq \mathbb{E}_{\boldsymbol{x}_{k-1}^n,\ldots,\boldsymbol{x}_1^0|\boldsymbol{x}_0}[\gamma V^{\boldsymbol{\pi}_n}(\boldsymbol{x}_{k-1}^n) + K] && \text{(Assumption 2.4)} \\
&= \gamma\mathbb{E}_{\boldsymbol{x}_{k-1}^n,\ldots,\boldsymbol{x}_1^0|\boldsymbol{x}_0}[V^{\boldsymbol{\pi}_n}(\boldsymbol{x}_{k-1}^n)] + K
\end{aligned}$$

We can apply Corollary A.6 to prove the claim. For the second claim, we note that for any $\boldsymbol{\pi}, \boldsymbol{\pi}'$ and $\boldsymbol{x} \in \mathcal{X}$ we have from Assumption 2.4

$$V^{\boldsymbol{\pi}}(\boldsymbol{x}) \leq C_u\alpha(\|\boldsymbol{x}\|) \leq \frac{C_u}{C_l}V^{\boldsymbol{\pi}'}(\boldsymbol{x}).$$

Therefore,

$$\begin{aligned}
&\mathbb{E}_{\boldsymbol{x}_0^n,\ldots,\boldsymbol{x}_1^0|\boldsymbol{x}_0}[V^{\boldsymbol{\pi}_n}(\boldsymbol{x}_0^n)] \\
&\leq \frac{C_u}{C_l}\mathbb{E}_{\boldsymbol{x}_0^n,\ldots,\boldsymbol{x}_1^0|\boldsymbol{x}_0}[V^{\boldsymbol{\pi}_{n-1}}(\boldsymbol{x}_0^n)] \\
&= \frac{C_u}{C_l}\mathbb{E}_{\boldsymbol{x}_{H_n}^{n-1},\ldots,\boldsymbol{x}_1^0|\boldsymbol{x}_0}[V^{\boldsymbol{\pi}_{n-1}}(\boldsymbol{x}_{H_n}^{n-1})] && \text{(Since } \boldsymbol{x}_0^n = \boldsymbol{x}_{H_n}^{n-1}\text{)} \\
&\leq \left(\frac{C_u}{C_l}\gamma^{H_n}\right)\mathbb{E}_{\boldsymbol{x}_0^{n-1},\ldots,\boldsymbol{x}_1^0|\boldsymbol{x}_0}[V^{\boldsymbol{\pi}_{n-1}}(\boldsymbol{x}_0^{n-1})] + \frac{C_u}{C_l}K/(1-\gamma) && \text{(Equation (12))}
\end{aligned}$$

For our choice of $H_0$, we have for all $n \geq 0$ that $\frac{C_u}{C_l}\gamma^{H_n} \leq \frac{C_u}{C_l}\gamma^{H_0} \leq \nu < 1$. From Corollary A.6, we get

$$\begin{aligned}
\mathbb{E}_{\boldsymbol{x}_0^n,\ldots,\boldsymbol{x}_1^0|\boldsymbol{x}_0}[V^{\boldsymbol{\pi}_n}(\boldsymbol{x}_0^n)] &\leq \left(\frac{C_u}{C_l}\gamma^{H_n}\right)\mathbb{E}_{\boldsymbol{x}_0^{n-1},\ldots,\boldsymbol{x}_1^0|\boldsymbol{x}_0}[V^{\boldsymbol{\pi}_{n-1}}(\boldsymbol{x}_0^{n-1})] + \frac{C_u}{C_l}K/(1-\gamma) \\
&\leq \nu\mathbb{E}_{\boldsymbol{x}_0^{n-1},\ldots,\boldsymbol{x}_1^0|\boldsymbol{x}_0}[V^{\boldsymbol{\pi}_{n-1}}(\boldsymbol{x}_0^{n-1})] + \frac{C_u}{C_l}K/(1-\gamma) \\
&\leq \nu^n V^{\boldsymbol{\pi}_0}(\boldsymbol{x}_0) + \frac{C_u}{C_l}K/(1-\gamma)\frac{1}{1-\nu}. && \text{(Corollary A.6)}
\end{aligned}$$

$$\begin{aligned}
\mathbb{E}_{\boldsymbol{x}_k^n,\ldots,\boldsymbol{x}_1^0|\boldsymbol{x}_0}[V^{\boldsymbol{\pi}_n}(\boldsymbol{x}_k^n)] &\leq \gamma^k\mathbb{E}_{\boldsymbol{x}_0^n,\ldots,\boldsymbol{x}_1^0|\boldsymbol{x}_0}[V^{\boldsymbol{\pi}_n}(\boldsymbol{x}_0^n)] + K/(1-\gamma) && \text{(Equation (12))} \\
&\leq \mathbb{E}_{\boldsymbol{x}_0^n,\ldots,\boldsymbol{x}_1^0|\boldsymbol{x}_0}[V^{\boldsymbol{\pi}_n}(\boldsymbol{x}_0^n)] + K/(1-\gamma) \\
&\leq \nu^n V^{\boldsymbol{\pi}_0}(\boldsymbol{x}_0) + \frac{C_u}{C_l}K/(1-\gamma)\frac{1}{1-\nu} + K/(1-\gamma) && \text{(Equation (13))} \\
&\leq V^{\boldsymbol{\pi}_0}(\boldsymbol{x}_0) + \frac{C_u}{C_l}K/(1-\gamma)\frac{1}{1-\nu} + K/(1-\gamma)
\end{aligned}$$

□

**Lemma A.8.** *Let Assumption 2.1 – 2.8 hold and let $H_0$ be the smallest integer such that*

$$H_0 > \frac{\log\left({C_u}/{C_l}\right)}{\log\left({1}/{\gamma}\right)}.$$

*Moreover, define $\nu = \frac{C_u}{C_l}\gamma^{H_0}$. Note, by definition of $H_0$, $\nu < 1$.*
*Then we have for all $k \in \{0, \ldots, H_n\}$ and $n > 0$*

Bounded second moment over horizon

$$\mathbb{E}_{\boldsymbol{x}_k^n,\ldots,\boldsymbol{x}_1^0|\boldsymbol{x}_0}\left[(V^{\boldsymbol{\pi}_n}(\boldsymbol{x}_k^n))^2\right] \leq \gamma^{2k}\mathbb{E}_{\boldsymbol{x}_0^n,\ldots,\boldsymbol{x}_1^0|\boldsymbol{x}_0}\left[(V^{\boldsymbol{\pi}_n}(\boldsymbol{x}_0^n))^2\right] + \frac{D_2(\boldsymbol{x}_0, K, \gamma, \nu)}{1-\gamma^2} \tag{15}$$

*with $D_2(\boldsymbol{x}_0, K, \gamma, \nu) = 2K\gamma D(\boldsymbol{x}_0, K, \gamma, \nu) + K^2 + C_{\boldsymbol{w}}$, and $C_{\boldsymbol{w}} = \mathbb{E}_{\boldsymbol{w}}\left[\kappa^2(\|w\|)\right] + 3(\mathbb{E}_{\boldsymbol{w}}\left[\kappa(\|w\|)\right])^2$.*

Bounded second moment over episodes

$$\mathbb{E}_{\boldsymbol{x}_0^n,\ldots,\boldsymbol{x}_1^0|\boldsymbol{x}_0}\left[\left(V^{\boldsymbol{\pi}_n}(\boldsymbol{x}_0^n)\right)^2\right] \leq \nu^{2n}\left(V^{\boldsymbol{\pi}_0}(\boldsymbol{x}_0)\right)^2 + \left(\frac{C_u}{C_l}\right)^2 \frac{D_2(\boldsymbol{x}_0,K,\gamma,\nu)}{1-\gamma^2}\frac{1}{1-\nu^2}. \tag{16}$$

*Moreover, let* $D_3(\boldsymbol{x}_0,K,\gamma,\nu) = \left(V^{\boldsymbol{\pi}_0}(\boldsymbol{x}_0)\right)^2 + D_2(\boldsymbol{x}_0,K,\gamma,\nu)\left(\left(\frac{C_u}{C_l}\right)^2\frac{1}{1-\gamma^2}\frac{1}{1-\nu^2}+\frac{1}{1-\gamma^2}\right)$.

$$\mathbb{E}_{\boldsymbol{x}_k^n,\ldots,\boldsymbol{x}_1^0|\boldsymbol{x}_0}\left[\left(V^{\boldsymbol{\pi}_n}(\boldsymbol{x}_k^n)\right)^2\right] \leq D_3(\boldsymbol{x}_0,K,\gamma,\nu)$$

*Proof.* Note that,

$$\begin{aligned}\mathbb{E}_{\boldsymbol{x}_k^n|\boldsymbol{x}_{k-1}^n}\left[\left(V^{\boldsymbol{\pi}_n}(\boldsymbol{x}_k^n)\right)^2\right] &= \left(\mathbb{E}_{\boldsymbol{x}_k^n|\boldsymbol{x}_{k-1}^n}\left[V^{\boldsymbol{\pi}_n}(\boldsymbol{x}_k^n)\right]\right)^2 \\ &\quad + \mathbb{E}_{\boldsymbol{x}_k^n|\boldsymbol{x}_{k-1}^n}\left[\left(V^{\boldsymbol{\pi}_n}(\boldsymbol{x}_k^n) - \mathbb{E}_{\boldsymbol{x}_k^n|\boldsymbol{x}_{k-1}^n}\left[V^{\boldsymbol{\pi}_n}(\boldsymbol{x}_k^n)\right]\right)^2\right].\end{aligned}$$

We first bound the second term. Let $\bar{\boldsymbol{x}}_k^n = \boldsymbol{f}^*(\boldsymbol{x}_{k-1}^n, \boldsymbol{\pi}_n(\boldsymbol{x}_{k-1}^n))$, i.e., the next state in the absence of transition noise.

$$\begin{aligned}&\mathbb{E}_{\boldsymbol{x}_k^n|\boldsymbol{x}_{k-1}^n}\left[\left(V^{\boldsymbol{\pi}_n}(\boldsymbol{x}_k^n) - \mathbb{E}_{\boldsymbol{x}_k^n|\boldsymbol{x}_{k-1}^n}\left[V^{\boldsymbol{\pi}_n}(\boldsymbol{x}_k^n)\right]\right)^2\right] \\ &\quad= \mathbb{E}_{\boldsymbol{x}_k^n|\boldsymbol{x}_{k-1}^n}\left[\left(V^{\boldsymbol{\pi}_n}(\boldsymbol{x}_k^n) - V^{\boldsymbol{\pi}_n}(\bar{\boldsymbol{x}}_k^n) + V^{\boldsymbol{\pi}_n}(\bar{\boldsymbol{x}}_k^n) - \mathbb{E}_{\boldsymbol{x}_k^n|\boldsymbol{x}_{k-1}^n}\left[V^{\boldsymbol{\pi}_n}(\boldsymbol{x}_k^n)\right]\right)^2\right] \\ &\quad= \mathbb{E}_{\boldsymbol{x}_k^n|\boldsymbol{x}_{k-1}^n}\left[\left(V^{\boldsymbol{\pi}_n}(\boldsymbol{x}_k^n) - V^{\boldsymbol{\pi}_n}(\bar{\boldsymbol{x}}_k^n) + \mathbb{E}_{\boldsymbol{x}_k^n|\boldsymbol{x}_{k-1}^n}\left[V^{\boldsymbol{\pi}_n}(\bar{\boldsymbol{x}}_k^n) - V^{\boldsymbol{\pi}_n}(\boldsymbol{x}_k^n)\right]\right)^2\right] \\ &\quad\leq \mathbb{E}_{\boldsymbol{w}}\left[\left(\kappa(\|w\|) + \mathbb{E}_{\boldsymbol{w}}[\kappa(\|w\|)]\right)^2\right] && \text{(uniform continuity of } V^{\boldsymbol{\pi}_n}) \\ &\quad= \mathbb{E}_{\boldsymbol{w}}\left[\kappa^2(\|\boldsymbol{w}\|)\right] + 3\left(\mathbb{E}_{\boldsymbol{w}}\left[\kappa(\|\boldsymbol{w}\|)\right]\right)^2 \\ &\quad= C_{\boldsymbol{w}} && \text{(Assumption 2.4)}\end{aligned}$$

Therefore we have

$$\begin{aligned}\mathbb{E}_{\boldsymbol{x}_k^n|\boldsymbol{x}_{k-1}^n}\left[\left(V^{\boldsymbol{\pi}_n}(\boldsymbol{x}_k^n)\right)^2\right] &= \left(\mathbb{E}_{\boldsymbol{x}_k^n|\boldsymbol{x}_{k-1}^n}\left[V^{\boldsymbol{\pi}_n}(\boldsymbol{x}_k^n)\right]\right)^2 + C_{\boldsymbol{w}} \\ &\leq \left(\gamma V^{\boldsymbol{\pi}_n}(\boldsymbol{x}_k^n) + K\right)^2 + C_{\boldsymbol{w}} \\ &= \gamma^2\left(V^{\boldsymbol{\pi}_n}(\boldsymbol{x}_{k-1}^n)\right)^2 + 2K\gamma V^{\boldsymbol{\pi}_n}(\boldsymbol{x}_{k-1}^n) + K^2 + C_{\boldsymbol{w}}.\end{aligned}$$

$$\begin{aligned}&\mathbb{E}_{\boldsymbol{x}_k^n,\ldots,\boldsymbol{x}_1^0|\boldsymbol{x}_0}\left[\left(V^{\boldsymbol{\pi}_n}(\boldsymbol{x}_k^n)\right)^2\right] \\ &= \mathbb{E}_{\boldsymbol{x}_{k-1}^n,\ldots,\boldsymbol{x}_1^0|\boldsymbol{x}_0}\left[\mathbb{E}_{\boldsymbol{x}_k^n|\boldsymbol{x}_{k-1}^n}\left[\left(V^{\boldsymbol{\pi}_n}(\boldsymbol{x}_k^n)\right)^2\right]\right] \\ &\leq \gamma^2\mathbb{E}_{\boldsymbol{x}_{k-1}^n,\ldots,\boldsymbol{x}_1^0|\boldsymbol{x}_0}\left[\left(V^{\boldsymbol{\pi}_n}(\boldsymbol{x}_{k-1}^n)\right)^2\right] + 2K\gamma\mathbb{E}_{\boldsymbol{x}_{k-1}^n,\ldots,\boldsymbol{x}_1^0|\boldsymbol{x}_0}\left[V^{\boldsymbol{\pi}_n}(\boldsymbol{x}_{k-1}^n)\right] + K^2 + C_{\boldsymbol{w}} \\ &\leq \gamma^2\mathbb{E}_{\boldsymbol{x}_{k-1}^n,\ldots,\boldsymbol{x}_1^0|\boldsymbol{x}_0}\left[\left(V^{\boldsymbol{\pi}_n}(\boldsymbol{x}_{k-1}^n)\right)^2\right] + 2K\gamma D(\boldsymbol{x}_0,K,\gamma,\nu) + K^2 + C_{\boldsymbol{w}}. && \text{(Lemma A.7)}\end{aligned}$$

Let $D_2(\boldsymbol{x}_0,K,\gamma,\nu) = 2K\gamma D(\boldsymbol{x}_0,K,\gamma,\nu) + K^2 + C_{\boldsymbol{w}}$. Applying Corollary A.6 we get

$$\mathbb{E}_{\boldsymbol{x}_k^n,\ldots,\boldsymbol{x}_1^0|\boldsymbol{x}_0}\left[\left(V^{\boldsymbol{\pi}_n}(\boldsymbol{x}_k^n)\right)^2\right] \leq \gamma^{2k}\mathbb{E}_{\boldsymbol{x}_0^n,\ldots,\boldsymbol{x}_1^0|\boldsymbol{x}_0}\left[\left(V^{\boldsymbol{\pi}_n}(\boldsymbol{x}_0^n)\right)^2\right] + \frac{D_2(\boldsymbol{x}_0,K,\gamma,\nu)}{1-\gamma^2}$$

Similar to the first moment, we leverage that $V^{\boldsymbol{\pi}_n}(\boldsymbol{x}) \leq \frac{C_u}{C_l}V^{\boldsymbol{\pi}_{n-1}}(\boldsymbol{x})$ for all $\boldsymbol{x} \in \mathcal{X}$, $\frac{C_u}{C_l}\gamma^{H_{n-1}} \leq \nu$, and get,

$$\mathbb{E}_{\boldsymbol{x}_0^n,\ldots,\boldsymbol{x}_1^0|\boldsymbol{x}_0}\left[\left(V^{\boldsymbol{\pi}_n}(\boldsymbol{x}_0^n)\right)^2\right]$$

$$
\begin{aligned}
&\leq \left(\frac{C_u}{C_l}\right)^2 \mathbb{E}_{\boldsymbol{x}_0^n, \dots, \boldsymbol{x}_1^0 | \boldsymbol{x}_0}\left[(V^{\boldsymbol{\pi}_{n-1}}(\boldsymbol{x}_0^n))^2\right] \\
&= \left(\frac{C_u}{C_l}\right)^2 \mathbb{E}_{\boldsymbol{x}_{H_n}^{n-1}, \dots, \boldsymbol{x}_1^0 | \boldsymbol{x}_0}\left[\left(V^{\boldsymbol{\pi}_{n-1}}(\boldsymbol{x}_{H_n}^{n-1})\right)^2\right] && \text{(Since } \boldsymbol{x}_0^n = \boldsymbol{x}_{H_n}^{n-1}\text{)} \\
&\leq \left(\frac{C_u}{C_l}\gamma^{H_n}\right)^2 \mathbb{E}_{\boldsymbol{x}_0^{n-1}, \dots, \boldsymbol{x}_1^0 | \boldsymbol{x}_0}\left[\left(V^{\boldsymbol{\pi}_{n-1}}(\boldsymbol{x}_0^{n-1})\right)^2\right] + \left(\frac{C_u}{C_l}\right)^2 \frac{D_2(\boldsymbol{x}_0, K, \gamma, \nu)}{1-\gamma^2} && \text{(Equation (15))} \\
&\leq \nu^2 \mathbb{E}_{\boldsymbol{x}_0^{n-1}, \dots, \boldsymbol{x}_1^0 | \boldsymbol{x}_0}\left[\left(V^{\boldsymbol{\pi}_{n-1}}(\boldsymbol{x}_0^{n-1})\right)^2\right] + \left(\frac{C_u}{C_l}\right)^2 \frac{D_2(\boldsymbol{x}_0, K, \gamma, \nu)}{1-\gamma^2} \\
&\leq \nu^{2n}\left(V^{\boldsymbol{\pi}_0}(\boldsymbol{x}_0)\right)^2 + \left(\frac{C_u}{C_l}\right)^2 \frac{D_2(\boldsymbol{x}_0, K, \gamma, \nu)}{1-\gamma^2}\frac{1}{1-\nu^2} && \text{(Corollary A.6)}
\end{aligned}
$$

Moreover,

$$
\begin{aligned}
&\mathbb{E}_{\boldsymbol{x}_k^n, \dots, \boldsymbol{x}_1^0 | \boldsymbol{x}_0}\left[(V^{\boldsymbol{\pi}_n}(\boldsymbol{x}_k^n))^2\right] \\
&\leq \gamma^{2k} \mathbb{E}_{\boldsymbol{x}_0^n, \dots, \boldsymbol{x}_1^0 | \boldsymbol{x}_0}\left[(V^{\boldsymbol{\pi}_n}(\boldsymbol{x}_0^n))^2\right] + \frac{D_2(\boldsymbol{x}_0, K, \gamma, \nu)}{1-\gamma^2} && \text{(Equation (15))} \\
&\leq \mathbb{E}_{\boldsymbol{x}_0^n, \dots, \boldsymbol{x}_1^0 | \boldsymbol{x}_0}\left[(V^{\boldsymbol{\pi}_n}(\boldsymbol{x}_0^n))^2\right] + \frac{D_2(\boldsymbol{x}_0, K, \gamma, \nu)}{1-\gamma^2} \\
&\leq \nu^{2n}\left(V^{\boldsymbol{\pi}_0}(\boldsymbol{x}_0)\right)^2 + \left(\frac{C_u}{C_l}\right)^2 \frac{D_2(\boldsymbol{x}_0, K, \gamma, \nu)}{1-\gamma^2}\frac{1}{1-\nu^2} + \frac{D_2(\boldsymbol{x}_0, K, \gamma, \nu)}{1-\gamma^2} && \text{(Equation (16))} \\
&\leq \left(V^{\boldsymbol{\pi}_0}(\boldsymbol{x}_0)\right)^2 + D_2(\boldsymbol{x}_0, K, \gamma, \nu)\left(\left(\frac{C_u}{C_l}\right)^2 \frac{1}{1-\gamma^2}\frac{1}{1-\nu^2} + \frac{1}{1-\gamma^2}\right)
\end{aligned}
$$

□

Finally, we prove the regret bound of NEORL.

*Proof of Theorem 3.1.* In the following, let $\hat{\boldsymbol{x}}_{k+1}^n = \boldsymbol{f}_n(\boldsymbol{x}_k^n, \boldsymbol{\pi}_n(\boldsymbol{x}_k^n)) + \boldsymbol{w}_k^n$ denote the state predicted under the optimistic dynamics and $\boldsymbol{x}_{k+1}^n = \boldsymbol{f}_n^*(\boldsymbol{x}_k^n, \boldsymbol{\pi}_n(\boldsymbol{x}_k^n)) + \boldsymbol{w}_k^n$ the true state.

$$
\begin{aligned}
&\mathbb{E}\left[\sum_{n=0}^{N-1}\sum_{k=0}^{H_n-1} c(\boldsymbol{x}_k^n, \boldsymbol{\pi}_n(\boldsymbol{x}_k^n)) - A(\boldsymbol{\pi}^*)\right] \\
&\leq \mathbb{E}\left[\sum_{n=0}^{N-1}\sum_{k=0}^{H_n-1} c(\boldsymbol{x}_k^n, \boldsymbol{\pi}_n(\boldsymbol{x}_k^n)) - A(\boldsymbol{\pi}_n, \boldsymbol{f}_n)\right] && \text{(Optimism)} \\
&= \mathbb{E}\left[\sum_{n=0}^{N-1}\sum_{k=0}^{H_n-1} B(\boldsymbol{\pi}_n, \boldsymbol{f}_n, \boldsymbol{x}_k^n) - B(\boldsymbol{\pi}_n, \boldsymbol{f}_n, \hat{\boldsymbol{x}}_{k+1}^n)\right] && \text{(Bellman equation ( Equation (4)))} \\
&= \mathbb{E}\left[\sum_{n=0}^{N-1}\sum_{k=0}^{H_n-1} B(\boldsymbol{\pi}_n, \boldsymbol{f}_n, \boldsymbol{x}_k^n) - B(\boldsymbol{\pi}_n, \boldsymbol{f}_n, \boldsymbol{x}_{k+1}^n) + B(\boldsymbol{\pi}_n, \boldsymbol{f}_n, \boldsymbol{x}_{k+1}^n) - B(\boldsymbol{\pi}_n, \boldsymbol{f}_n, \hat{\boldsymbol{x}}_{k+1}^n)\right] \\
&= \sum_{n=0}^{N-1}\sum_{k=0}^{H_n-1} \mathbb{E}\left[B(\boldsymbol{\pi}_n, \boldsymbol{f}_n, \boldsymbol{x}_{k+1}^n) - B(\boldsymbol{\pi}_n, \boldsymbol{f}_n, \hat{\boldsymbol{x}}_{k+1}^n)\right] && \text{(A)} \\
&+ \sum_{n=0}^{N-1}\sum_{k=0}^{H_n-1} \mathbb{E}\left[B(\boldsymbol{\pi}_n, \boldsymbol{f}_n, \boldsymbol{x}_k^n) - B(\boldsymbol{\pi}_n, \boldsymbol{f}_n, \boldsymbol{x}_{k+1}^n)\right] && \text{(B)}
\end{aligned}
$$

First, we study the term (A).

**Proof for (A)**: Note that because $\boldsymbol{f}_n \in \mathcal{M}_n$, there exists a $\boldsymbol{\eta} \in [-1, 1]^{d_x}$ such that $\hat{\boldsymbol{x}}_{k+1}^n = \boldsymbol{\mu}_n(\boldsymbol{x}_k^n, \boldsymbol{\pi}_n(\boldsymbol{x}_k^n)) + \beta_n \boldsymbol{\sigma}_n(\boldsymbol{x}_k^n, \boldsymbol{\pi}_n(\boldsymbol{x}_k^n))\boldsymbol{\eta}(\boldsymbol{x}_k^n) + \boldsymbol{w}_k^n$. Furthermore, $\boldsymbol{x}_{k+1}^n = \boldsymbol{f}^*(\boldsymbol{x}_k^n, \boldsymbol{\pi}_n(\boldsymbol{x}_k^n)) + \boldsymbol{w}_k^n$ and the transition noise is Gaussian. Let $\zeta_{2,k}^n$ and $\zeta_{1,k}^n$ denote the respective distributions of the

two random variables, i.e., $\zeta_{1,k}^n \sim \mathcal{N}(\boldsymbol{f}^*(\boldsymbol{x}_k^n, \boldsymbol{\pi}_n(\boldsymbol{x}_k^n)), \sigma^2 \boldsymbol{I})$ and $\zeta_{2,k}^n \sim \mathcal{N}(\boldsymbol{f}_n(\boldsymbol{x}_k^n, \boldsymbol{\pi}_n(\boldsymbol{x}_k^n)), \sigma^2 \boldsymbol{I})$. Next, define $\bar{B} = \mathbb{E}_{\boldsymbol{x} \sim \zeta_{2,k}^n}[B(\boldsymbol{\pi}_n, \boldsymbol{f}_n, \boldsymbol{x})]$, and consider the function $h(\boldsymbol{x}) = B(\boldsymbol{\pi}_n, \boldsymbol{f}_n, \boldsymbol{x}) - \bar{B}$. Then we have

$$\begin{aligned}
&\mathbb{E}_{\boldsymbol{w}_k^n}\left[B(\boldsymbol{\pi}_n, \boldsymbol{f}_n, \boldsymbol{x}_{k+1}^n) - B(\boldsymbol{\pi}_n, \boldsymbol{f}_n, \hat{\boldsymbol{x}}_{k+1}^n)\right] \\
&\quad = \mathbb{E}_{\boldsymbol{x} \sim \zeta_{1,k}^n}\left[B(\boldsymbol{\pi}_n, \boldsymbol{f}_n, \boldsymbol{x})\right] - \mathbb{E}_{\boldsymbol{x} \sim \zeta_{2,k}^n}\left[B(\boldsymbol{\pi}_n, \boldsymbol{f}_n, \boldsymbol{x})\right] \\
&\quad = \mathbb{E}_{\boldsymbol{x} \sim \zeta_{1,k}^n}\left[B(\boldsymbol{\pi}_n, \boldsymbol{f}_n, \boldsymbol{x}) - \bar{B}\right] - \mathbb{E}_{\boldsymbol{x} \sim \zeta_{2,k}^n}\left[B(\boldsymbol{\pi}_n, \boldsymbol{f}_n, \boldsymbol{x}) - \bar{B}\right] \\
&\quad = \mathbb{E}_{\boldsymbol{x} \sim \zeta_{1,k}^n}[h(\boldsymbol{x})] - \mathbb{E}_{\boldsymbol{x} \sim \zeta_{2,k}^n}[h(\boldsymbol{x})].
\end{aligned}$$

Note that $\mathbb{E}_{\boldsymbol{x} \sim \zeta_{2,k}^n}[h(\boldsymbol{x})] = 0$ by the definition of $h$ and thus,

$$\mathbb{E}_{\boldsymbol{x} \sim \zeta_{1,k}^n}[h(\boldsymbol{x})] - \mathbb{E}_{\boldsymbol{x} \sim \zeta_{2,k}^n}[h(\boldsymbol{x})] = \mathbb{E}_{\boldsymbol{x} \sim \zeta_{1,k}^n}[h(\boldsymbol{x})] \leq \sqrt{\mathbb{E}_{\boldsymbol{x} \sim \zeta_{1,k}^n}[h^2(\boldsymbol{x})]}. \tag{17}$$

In the following, we bound the term above w.r.t. the Chi-squared distance

$$\begin{aligned}
&\mathbb{E}_{\boldsymbol{w}_k^n}\left[B(\boldsymbol{\pi}_n, \boldsymbol{f}_n, \boldsymbol{x}_{k+1}^n) - B(\boldsymbol{\pi}_n, \boldsymbol{f}_n, \hat{\boldsymbol{x}}_{k+1}^n)\right] = \mathbb{E}_{\boldsymbol{x} \sim \zeta_{1,k}^n}[h(\boldsymbol{x})] - \mathbb{E}_{\boldsymbol{x} \sim \zeta_{2,k}^n}[h(\boldsymbol{x})] \\
&= \int_{\mathcal{X}} h(\boldsymbol{x}) \left(1 - \frac{\zeta_{2,k}^n}{\zeta_{1,k}^n}\right) \zeta_{1,k}^n(d\boldsymbol{x}) \leq \sqrt{\mathbb{E}_{\boldsymbol{x} \sim \zeta_{1,k}^n}\left[h^2(\boldsymbol{x})\right]} \sqrt{d_{\mathcal{X}}(\zeta_{2,k}^n, \zeta_{1,k}^n)}
\end{aligned}$$

((Kakade et al., 2020, Lemma C.2.,))

With $d_{\mathcal{X}}(\zeta_{2,k}^n, \zeta_{1,k}^n)$ being the Chi-squared distance.

$$d_{\mathcal{X}}(\zeta_{2,k}^n, \zeta_{1,k}^n) = \int_{\mathcal{X}} \frac{\left(\zeta_{1,k}^n - \zeta_{2,k}^n\right)^2}{\zeta_{1,k}^n}(d\boldsymbol{x})$$

Since both bounds from Equation (17) and bound we got by applying (Kakade et al., 2020, Lemma C.2.,), we can apply minimum and have:

$$\mathbb{E}_{\boldsymbol{w}_k^n}\left[B(\boldsymbol{\pi}_n, \boldsymbol{f}_n, \boldsymbol{x}_{k+1}^n) - B(\boldsymbol{\pi}_n, \boldsymbol{f}_n, \hat{\boldsymbol{x}}_{k+1}^n)\right] \leq \sqrt{\mathbb{E}_{\boldsymbol{x} \sim \zeta_{1,k}^n}\left[h^2(\boldsymbol{x})\right]} \sqrt{\min\left\{d_{\mathcal{X}}(\zeta_{2,k}^n, \zeta_{1,k}^n), 1\right\}}$$

Therefore, following Kakade et al. (2020, Lemma C.2.,) we get

$$\begin{aligned}
&\mathbb{E}_{\boldsymbol{w}_k^n}\left[B(\boldsymbol{\pi}_n, \boldsymbol{f}_n, \boldsymbol{x}_{k+1}^n) - B(\boldsymbol{\pi}_n, \boldsymbol{f}_n, \hat{\boldsymbol{x}}_{k+1}^n)\right] \\
&\quad \leq \sqrt{\mathbb{E}_{\boldsymbol{x} \sim \zeta_{1,k}^n}\left[h^2(\boldsymbol{x})\right]} \min\left\{{}^1\!/\!_\sigma \left\|\boldsymbol{f}^*(\boldsymbol{x}_k^n, \boldsymbol{\pi}_n(\boldsymbol{x}_k^n)) - \boldsymbol{f}_n(\boldsymbol{x}_k^n, \boldsymbol{\pi}_n(\boldsymbol{x}_k^n))\right\|, 1\right\} \\
&\quad \leq \sqrt{\mathbb{E}_{\boldsymbol{x} \sim \zeta_{1,k}^n}\left[h^2(\boldsymbol{x})\right]} (1 + \sqrt{d}_x)^{\beta_n/\sigma} \left\|\boldsymbol{\sigma}_n(\boldsymbol{x}_k^n, \boldsymbol{\pi}_n(\boldsymbol{x}_k^n))\right\|. \qquad \text{((Sukhija et al., 2024, Cor. 3))}
\end{aligned}$$

Therefore, we have

$$\begin{aligned}
&\sum_{n=0}^{N-1} \sum_{k=0}^{H_n-1} \mathbb{E}_{\boldsymbol{x}_k^n, \ldots \boldsymbol{x}_1^0 | \boldsymbol{x}_0}\left[\mathbb{E}_{\boldsymbol{w}_k^n}\left[B(\boldsymbol{\pi}_n, \boldsymbol{f}_n, \boldsymbol{x}_{k+1}^n) - B(\boldsymbol{\pi}_n, \boldsymbol{f}_n, \hat{\boldsymbol{x}}_{k+1}^n)\right]\right] \\
&\leq \sum_{n=0}^{N-1} \sum_{k=0}^{H_n-1} \mathbb{E}_{\boldsymbol{x}_k^n, \ldots \boldsymbol{x}_1^0 | \boldsymbol{x}_0}\left[\sqrt{\mathbb{E}_{\boldsymbol{x} \sim \zeta_{1,k}^n}\left[h^2(\boldsymbol{x})\right]} (1 + \sqrt{d}_x)^{\beta_n/\sigma} \left\|\boldsymbol{\sigma}_n(\boldsymbol{x}_k^n, \boldsymbol{\pi}_n(\boldsymbol{x}_k^n))\right\|\right] \\
&\leq \sum_{n=0}^{N-1} \sum_{k=0}^{H_n-1} (1 + \sqrt{d}_x)^{\beta_n/\sigma} \sqrt{\mathbb{E}_{\boldsymbol{x}_k^n, \ldots \boldsymbol{x}_1^0 | \boldsymbol{x}_0}\left[\mathbb{E}_{\boldsymbol{x} \sim \zeta_{1,k}^n}\left[h^2(\boldsymbol{x})\right]\right] \mathbb{E}_{\boldsymbol{x}_k^n, \ldots \boldsymbol{x}_1^0 | \boldsymbol{x}_0}\left[\left\|\boldsymbol{\sigma}_n(\boldsymbol{x}_k^n, \boldsymbol{\pi}_n(\boldsymbol{x}_k^n))\right\|^2\right]} \\
&\leq (1 + \sqrt{d}_x)^{\beta_T/\sigma} \sqrt{\sum_{n=0}^{N-1} \sum_{k=0}^{H_n-1} \mathbb{E}_{\boldsymbol{x}_k^n, \ldots \boldsymbol{x}_1^0 | \boldsymbol{x}_0}\left[\mathbb{E}_{\boldsymbol{x} \sim \zeta_{1,k}^n}\left[h^2(\boldsymbol{x})\right]\right]} \\
&\times \sqrt{\sum_{n=0}^{N-1} \sum_{k=0}^{H_n-1} \mathbb{E}_{\boldsymbol{x}_k^n, \ldots \boldsymbol{x}_1^0 | \boldsymbol{x}_0}\left[\left\|\boldsymbol{\sigma}_n(\boldsymbol{x}_k^n, \boldsymbol{\pi}_n(\boldsymbol{x}_k^n))\right\|^2\right]}
\end{aligned}$$

Here, for the second and third inequality, we use Cauchy-Schwarz. Now we bound the two terms above individually.
First we bound $\mathbb{E}_{\boldsymbol{x}\sim\zeta_{1,k}^n}\left[h^2(\boldsymbol{x})\right]$.

$$\begin{aligned}
\mathbb{E}_{\boldsymbol{x}\sim\zeta_{1,k}^n}\left[h^2(\boldsymbol{x})\right] &= \mathbb{E}_{\boldsymbol{x}\sim\zeta_{1,k}^n}\left[(B(\boldsymbol{\pi}_n,\boldsymbol{f}_n,\boldsymbol{x})-\bar{B})^2\right] \\
&= \mathbb{E}_{\boldsymbol{x}\sim\zeta_{1,k}^n}\left[(B(\boldsymbol{\pi}_n,\boldsymbol{f}_n,\boldsymbol{x})-\mathbb{E}_{\boldsymbol{x}\sim\zeta_{2,k}^n}\left[B(\boldsymbol{\pi}_n,\boldsymbol{f}_n,\boldsymbol{x})\right])^2\right] \\
&\leq \left(\frac{C_2}{1-\hat{\lambda}}\right)^2 \mathbb{E}_{\boldsymbol{x}\sim\zeta_{1,k}^n}\left[(2+V^{\boldsymbol{\pi}_n}(\boldsymbol{x})+\mathbb{E}_{\boldsymbol{x}\sim\zeta_{2,k}^n}\left[V^{\boldsymbol{\pi}_n}(\boldsymbol{x})\right])^2\right] && \text{(Theorem A.3)} \\
&\leq \left(\frac{C_2}{1-\hat{\lambda}}\right)^2 \mathbb{E}_{\boldsymbol{x}\sim\zeta_{1,k}^n}\left[(2+V^{\boldsymbol{\pi}_n}(\boldsymbol{x})+\gamma V^{\boldsymbol{\pi}_n}(\boldsymbol{x}_k^n)+\hat{K})^2\right] && \text{(Lemma A.4)} \\
&\leq \left(\frac{\sqrt{2}C_2}{1-\hat{\lambda}}\right)^2 \mathbb{E}_{\boldsymbol{x}\sim\zeta_{1,k}^n}\left[(V^{\boldsymbol{\pi}_n}(\boldsymbol{x}))^2+(2+\gamma V^{\boldsymbol{\pi}_n}(\boldsymbol{x}_k^n)+\hat{K})^2\right] \\
&\leq \left(\frac{\sqrt{2}C_2}{1-\hat{\lambda}}\right)^2 \left(\mathbb{E}_{\boldsymbol{x}_{k+1}^n|\boldsymbol{x}_k^n}\left[(V^{\boldsymbol{\pi}_n}(\boldsymbol{x}_{k+1}))^2\right]+2\gamma^2(V^{\boldsymbol{\pi}_n}(\boldsymbol{x}_k^n))^2+2(2+\hat{K})^2\right)
\end{aligned}$$

Furthermore, we have from Lemma A.8.

$$\begin{aligned}
&\mathbb{E}_{\boldsymbol{x}_k^n,\ldots\boldsymbol{x}_1^0|\boldsymbol{x}_0}\left[\mathbb{E}_{\boldsymbol{x}_{k+1}^n|\boldsymbol{x}_k^n}\left[(V^{\boldsymbol{\pi}_n}(\boldsymbol{x}_{k+1}))^2\right]+2\gamma^2(V^{\boldsymbol{\pi}_n}(\boldsymbol{x}_k^n))^2\right] \\
&= \mathbb{E}_{\boldsymbol{x}_{k+1}^n,\ldots\boldsymbol{x}_1^0|\boldsymbol{x}_0}\left[(V^{\boldsymbol{\pi}_n}(\boldsymbol{x}_{k+1}))^2\right]+2\gamma^2\mathbb{E}_{\boldsymbol{x}_k^n,\ldots\boldsymbol{x}_1^0|\boldsymbol{x}_0}\left[(V^{\boldsymbol{\pi}_n}(\boldsymbol{x}_k^n))^2\right] \leq (1+2\gamma^2)D_3(\boldsymbol{x}_0,K,\gamma,\nu).
\end{aligned}$$

In the end, we get

$$\begin{aligned}
&\sqrt{\sum_{n=0}^{N-1}\sum_{k=0}^{H_n-1}\mathbb{E}_{\boldsymbol{x}_k^n,\ldots\boldsymbol{x}_1^0|\boldsymbol{x}_0}\left[\mathbb{E}_{\boldsymbol{x}\sim\zeta_{1,k}^n}\left[h^2(\boldsymbol{x})\right]\right]} \\
&\leq \left(\frac{\sqrt{2}C_2}{1-\hat{\lambda}}\right)\sqrt{\sum_{n=0}^{N-1}\sum_{k=0}^{H_n-1}(1+2\gamma^2)D_3(\boldsymbol{x}_0,K,\gamma,\nu)+2(2+\hat{K})^2} \\
&= \left(\frac{\sqrt{2}C_2}{1-\hat{\lambda}}\right)\sqrt{(1+2\gamma^2)D_3(\boldsymbol{x}_0,K,\gamma,\nu)+2(2+\hat{K})^2}\sqrt{\sum_{n=0}^{N-1}H_n} \\
&= \left(\frac{\sqrt{2}C_2}{1-\hat{\lambda}}\right)\sqrt{(1+2\gamma^2)D_3(\boldsymbol{x}_0,K,\gamma,\nu)+2(2+\hat{K})^2}\sqrt{T}.
\end{aligned}$$

Next, we use the bound from Curi et al. (2020, Lemma 17.) for the second term.

$$\sqrt{\sum_{n=0}^{N-1}\sum_{k=0}^{H_n-1}\mathbb{E}_{\boldsymbol{x}_k^n,\ldots\boldsymbol{x}_1^0|\boldsymbol{x}_0}\left[\|\boldsymbol{\sigma}_n(\boldsymbol{x}_k^n,\boldsymbol{\pi}_n(\boldsymbol{x}_k^n))\|^2\right]} \leq C'\sqrt{\Gamma_T}$$

Here $\Gamma_T$ is the maximum information gain.
If we set $D_4(\boldsymbol{x}_0,K,\gamma) = \frac{C'(1+\sqrt{d_x})}{\sigma}\left(\frac{\sqrt{2}C_2}{1-\hat{\lambda}}\right)\sqrt{(1+2\gamma^2)D_3(\boldsymbol{x}_0,K,\gamma,\nu)+2(2+\hat{K})^2}$, we have

$$\begin{aligned}
&\sum_{n=0}^{N-1}\sum_{k=0}^{H_n-1}\mathbb{E}_{\boldsymbol{x}_k^n,\ldots\boldsymbol{x}_1^0|\boldsymbol{x}_0}\left[\mathbb{E}_{\boldsymbol{w}_k^n}\left[B(\boldsymbol{\pi}_n,\boldsymbol{f}_n,\boldsymbol{x}_{k+1}^n)-B(\boldsymbol{\pi}_n,\boldsymbol{f}_n,\hat{\boldsymbol{x}}_{k+1}^n)\right]\right] \\
&\leq (1+\sqrt{d_x})^{\beta_T/\sigma}\sqrt{\sum_{n=0}^{N-1}\sum_{k=0}^{H_n-1}\mathbb{E}_{\boldsymbol{x}_k^n,\ldots\boldsymbol{x}_1^0|\boldsymbol{x}_0}\left[\mathbb{E}_{\boldsymbol{x}\sim\zeta_{1,k}^n}\left[h^2(\boldsymbol{x})\right]\right]}
\end{aligned}$$

$$
\begin{aligned}
&\times \sqrt{\sum_{n=0}^{N-1} \sum_{k=0}^{H_n-1} \mathbb{E}_{\boldsymbol{x}_k^n,\dots \boldsymbol{x}_1^0 | \boldsymbol{x}_0} \left[ \left\| \boldsymbol{\sigma}_n(\boldsymbol{x}_k^n, \boldsymbol{\pi}_n(\boldsymbol{x}_k^n)) \right\|^2 \right]} \\
&\leq (1+\sqrt{d_x})^{\beta_T/\sigma} \left( \frac{\sqrt{2}C_2}{1-\hat{\lambda}} \right) \sqrt{(1+2\gamma^2) D_3(\boldsymbol{x}_0, K, \gamma, \nu) + 2(2+\hat{K})^2} \sqrt{T} C' \sqrt{\Gamma_T} \\
&\leq D_4(\boldsymbol{x}_0, K, \gamma) \beta_T \sqrt{T \Gamma_T}
\end{aligned}
$$

**Proof for (B)**:

$$
\begin{aligned}
&\sum_{n=0}^{N-1} \sum_{k=0}^{H_n-1} \mathbb{E}\left[ B(\boldsymbol{\pi}, \boldsymbol{f}_n, \boldsymbol{x}_k^n) - B(\boldsymbol{\pi}, \boldsymbol{f}_n, \boldsymbol{x}_{k+1}^n) \right] = \sum_{n=0}^{N-1} \mathbb{E}\left[ B(\boldsymbol{\pi}, \boldsymbol{f}_n, \boldsymbol{x}_0^n) - B(\boldsymbol{\pi}, \boldsymbol{f}_n, \boldsymbol{x}_{H_n}^n) \right] \\
&\quad \leq \frac{C_2}{1-\hat{\lambda}} \sum_{n=0}^{N-1} \left( 2 + \mathbb{E}\left[ V^{\boldsymbol{\pi}}(\boldsymbol{x}_0^n) + V^{\boldsymbol{\pi}}(\boldsymbol{x}_{H_n}^n) \right] \right) && \text{(Theorem A.3)} \\
&\quad \leq \frac{2C_2}{1-\hat{\lambda}} \sum_{n=0}^{N-1} (1 + D(\boldsymbol{x}_0, K, \gamma)) && \text{(Lemma A.7)} \\
&\quad = \frac{2C_2}{1-\hat{\lambda}} (1 + D(\boldsymbol{x}_0, K, \gamma)) N \\
&\quad = D_5(\boldsymbol{x}_0, K, \gamma) N.
\end{aligned}
$$

Here $D_5(\boldsymbol{x}_0, K, \gamma) = \frac{2C_2}{1-\hat{\lambda}}(1 + D(\boldsymbol{x}_0, K, \gamma))$. Finally, for our choice, $H_n = H_0 2^n$, we get

$$
\sum_{n=0}^{N-1} H_n = H_0 \sum_{n=0}^{N-1} 2^n = H_0(2^N - 1) = T.
$$

Therefore, $N = \log_2 \left( \frac{T}{H_0} + 1 \right)$. To this end, we get for our regret

$$
\begin{aligned}
R_T &= \mathbb{E}\left[ \sum_{n=0}^{N-1} \sum_{k=0}^{H_n-1} c(\boldsymbol{x}_k^n, \boldsymbol{\pi}_n(\boldsymbol{x}_k^n)) - A(\boldsymbol{\pi}^*) \right] \\
&\leq D_4(\boldsymbol{x}_0, K, \gamma) \beta_T \sqrt{T \Gamma_T} + D_5(\boldsymbol{x}_0, K, \gamma) N \\
&\leq D_4(\boldsymbol{x}_0, K, \gamma) \beta_T \sqrt{T \Gamma_T} + D_5(\boldsymbol{x}_0, K, \gamma) \log_2 \left( \frac{T}{H_0} + 1 \right)
\end{aligned}
$$

□

This regret is sublinear for a very rich class of functions. We summarize bounds on $\Gamma_T$ from Vakili et al. (2021) in Table 1. Furthermore, note that $D_4(\boldsymbol{x}_0, K, \gamma) \in (0, \infty)$ for all $\boldsymbol{x}_0 \in \mathcal{X}$ with $\|\boldsymbol{x}_0\| < \infty$, $K < \infty$, $\gamma \in (0, 1)$. The same holds for $D_5(\boldsymbol{x}_0, K, \gamma)$. Moreover, since $V^{\boldsymbol{\pi}}(\boldsymbol{x})$ is $\Theta(\zeta(\|\boldsymbol{x}\|))$, both $D_4$ and $D_5$ are $\Theta(\zeta(\|\boldsymbol{x}_0\|))$.

Table 1: Maximum information gain bounds for common choice of kernels.

| Kernel | $k(\boldsymbol{x}, \boldsymbol{x}')$ | $\Gamma_T$ |
|---|---|---|
| Linear | $\boldsymbol{x}^\top \boldsymbol{x}'$ | $\mathcal{O}\left(d \log(T)\right)$ |
| RBF | $e^{-\frac{\|\boldsymbol{x}-\boldsymbol{x}'\|^2}{2l^2}}$ | $\mathcal{O}\left(\log^{d+1}(T)\right)$ |
| Matèrn | $\frac{1}{\Gamma(\nu)2^{\nu-1}} \left( \frac{\sqrt{2\nu}\|\boldsymbol{x}-\boldsymbol{x}'\|}{l} \right)^\nu B_\nu \left( \frac{\sqrt{2\nu}\|\boldsymbol{x}-\boldsymbol{x}'\|}{l} \right)$ | $\mathcal{O}\left(T^{\frac{d}{2\nu+d}} \log^{\frac{2\nu}{2\nu+d}}(T)\right)$ |

**Algorithm 2** Practical NEORL:

**Init:** Aleatoric uncertainty $\sigma$, Probability $\delta$, Statistical model $(\boldsymbol{\mu}_0, \boldsymbol{\sigma}_0, \beta_0(\delta))$
**for** $n = 1, \ldots, N$ **do**
  **for** $h = 1, \ldots, H$ **do**

$$\min_{\boldsymbol{u}_{0:H_{\text{MPC}}-1}, \boldsymbol{\eta}_{0:H_{\text{MPC}}-1}} \mathbb{E}\left[\sum_{h=0}^{H_{\text{MPC}}-1} c(\hat{\boldsymbol{x}}_h, \boldsymbol{u}_h)\right]; \boldsymbol{x}_0 = \boldsymbol{x}_h^n$$ ➤ Solve MPC problem

$(\boldsymbol{x}_n^h, \boldsymbol{u}_0^*, \boldsymbol{x}_n^{h+1}) \leftarrow \text{ROLLOUT}(\boldsymbol{u}_0^*)$ ➤ Collect transition

  **end for**
  Update $(\boldsymbol{\mu}_n, \boldsymbol{\sigma}_n, \beta_n) \leftarrow \mathcal{D}_n$
**end for**

# B Practical algorithm and Experimental Details

In this section, we provide the practical algorithm Algorithm 2, provide all hyperparameters used in our experiments in Table 2, and the cost function for the environments. All our experiments within 1-8 hours[3] on a GPU (NVIDIA GeForce RTX 2080 Ti). For NEORL, we use $\beta_n = 2$ for all the experiments, except for the Swimmer and the SoftArm environment where we use $\beta_n = 1$.

Table 2: Hyperparameters for results in Section 4.

| Environment | iCEM parameters | | | | | Model training parameters | | | | | | |
|---|---|---|---|---|---|---|---|---|---|---|---|---|
| | Number of samples | Number of elites | Optimizer steps | $H_{\text{MPC}}$ | Particles | Number of ensembles | Network architecture | Learning rate | Batch size | Number of epochs | H | Action Repeat |
| Pendulum-GP | 500 | 50 | 10 | 20 | 5 | - | - | 0.01 | 64 | - | 10 | 1 |
| Pendulum | 500 | 50 | 10 | 20 | 5 | 10 | $256 \times 2$ | 0.001 | 64 | 50 | 10 | 1 |
| MountainCar | 1000 | 100 | 5 | 50 | 5 | 10 | $256 \times 2$ | 0.001 | 64 | 50 | 10 | 2 |
| Reacher | 1000 | 100 | 10 | 50 | 5 | 10 | $256 \times 2$ | 0.001 | 64 | 50 | 10 | 2 |
| CartPole | 1000 | 100 | 10 | 50 | 5 | 10 | $256 \times 2$ | 0.001 | 64 | 50 | 10 | 2 |
| Swimmer | 500 | 50 | 10 | 30 | 5 | 10 | $256 \times 4$ | 0.00005 | 64 | 100 | 200 | 4 |
| SoftArm | 500 | 50 | 10 | 20 | 5 | 10 | $256 \times 4$ | 0.00005 | 64 | 50 | 20 | 1 |
| RaceCar | 1000 | 100 | 10 | 50 | 5 | 10 | $256 \times 2$ | 0.001 | 64 | 50 | 10 | 1 |

Table 3: Cost function for the environments presented in Section 4.

| **Environment** | **Cost** $c(\boldsymbol{x}_t, \boldsymbol{u}_t)$ |
|---|---|
| Pendulum | $\theta_t^2 + 0.1\dot{\theta}_t + 0.1u_t^2$ |
| MountainCar | $0.1u_t^2 + 100(1\{\boldsymbol{x}_t \notin \boldsymbol{x}_{\text{goal}}\})$ |
| Reacher | $\|\boldsymbol{x}_t - \boldsymbol{x}_{\text{target}}\| + 0.1\,\|u_t\|$ |
| CartPole | $\left\|\boldsymbol{x}_t^{\text{pos}} - \boldsymbol{x}_{\text{target}}^{\text{pos}}\right\|^2 + 10(\cos(\theta_t) - 1)^2 + 0.2\,\|u_t\|^2$ |
| Swimmer | $\|\boldsymbol{x}_t - \boldsymbol{x}_{\text{target}}\|$ |
| SoftArm | $\|\boldsymbol{x}_t - \boldsymbol{x}_{\text{target}}\|$ |
| RaceCar | $\|\boldsymbol{x}_t - \boldsymbol{x}_{\text{target}}\|$ |

[3]based on the environment

