# OpenReview forum: "NeoRL: Efficient Exploration for Nonepisodic RL"
_NeurIPS.cc/2024/Conference — NeurIPS 2024 spotlight_

### Official Review · Reviewer_WMig · 2024-07-11

**Soundness:** 4
**Presentation:** 3
**Contribution:** 3
**Rating:** 7
**Confidence:** 2

**Summary:**

This paper proposes a model-based RL algorithm NeoRL  for continuous state-action spaces in the nonepisodic setting, where the agent learns from a single trajectory without resets.

**Strengths:**

1. The paper provides the first regret bound for nonepisodic RL in general nonlinear systems, addressing a gap in the literature.
2. NeoRL is grounded in the optimism principle and leverages well-calibrated probabilistic models, providing a theoretically justified exploration strategy.
3. The experiments demonstrate the practical effectiveness of NEORL, achieving sublinear regret and converging to the optimal average cost in various environments.

**Weaknesses:**

1. The optimization problem in Equation (6) for policy selection may be computationally expensive, especially for high-dimensional systems.
2. While the experiments demonstrate the efficacy of NEORL, a more comprehensive evaluation across a diverse set of environments and comparisons with state-of-the-art methods would strengthen the empirical results.
3. The authors can discuss some data-driven MPC framework, like [1,2].

[1] Berberich J, Köhler J, Müller M A, et al. Data-driven model predictive control with stability and robustness guarantees[J]. IEEE Transactions on Automatic Control, 2020, 66(4): 1702-1717.

[2] Berberich J, Allgöwer F. An overview of systems-theoretic guarantees in data-driven model predictive control[J]. arXiv preprint arXiv:2406.04130, 2024.

**Questions:**

1. How sensitive is the performance of NeoRL to the choice of kernel and the maximum information gain? What guidance can be provided for selecting appropriate kernels in practice?
2. Can the bounded energy assumption be relaxed or replaced with alternative assumptions to broaden the applicability of NEORL?

**Limitations:**

The authors have discussed the limitations.

---

> ### Author Rebuttal · Authors · 2024-08-05
>
> We thank the reviewer for their feedback and are happy to see the reviewer acknowledge how our work bridges a significant gap in RL theory. In the following, we respond to the weaknesses and questions raised by the reviewer.
>
> **W1**: Expensive computation.
>
> **A1**: Thanks for raising this concern. Indeed the optimization problem in (6) may fair infavourably for high-dimensional systems. However, there are heuristic approximations that can be made that favor much better for the high dimensional case (c.f., [1, 2] or our response to reviewer zoj1’s second question).
>
> **W2**: Comparisons with state-of-the-art methods.
>
> **A2**:  In our experimental evaluation, we focus only on model-based methods due to their sample efficiency. As we highlight in section 4, MBRL methods vary mostly in how the model is represented (GPs, BNNs, world models, etc), the choice of policy optimizer, and how the dynamics are propagated to facilitate exploration. Our contribution is for the third axis of differentiation, i.e., we show that the celebrated principle of optimism works for many cases in the nonepisodic setting. Thereby we compare our method to other approaches of dynamics propagation such as mean (which is most widely used), trajectory sampling (PETS [3]), and Thompson sampling. To the best of our knowledge, these are the only approaches (along the third axis) typically considered in MBRL.
>
> **W3**: Discuss some data-driven MPC framework.
>
> **A3**: Thanks for pointing us to these references, we have added them to our related work.
>
>
> **Q1**: Sensitivity to the choice of kernel.
>
> **A1**: In theory, the choice of the kernel has an effect on the convergence guarantees/max info gain (c.f., Table 1 in Appendix A). Practically, we used the most common kernel, the Gaussian kernel, in our experiments which worked completely fine. However, according to the theory, choosing the right kernel can affect the performance/convergence of the underlying algorithm. One way to make this choice is to use some offline/pre-recorded data on the system for kernel selection or meta-learn the kernel parameters from other data sources (c.f., Rothfuss, Jonas, et al. "Meta-learning priors for safe Bayesian optimization." Conference on Robot Learning. PMLR, 2023)
>
> **Q2**:  Bounded energy assumption.
>
> **A2**: Thanks for this very interesting question. The bounded energy assumption stems from the classical results on Markov chains (Meyn, Sean P., and Richard L. Tweedie. Markov chains and stochastic stability. Springer Science & Business Media, 2012.), therefore we are unsure if this can be further relaxed. Nonetheless, we are currently looking into possible ways to relax this assumption. However, as highlighted in Corollary 2.5., for many practical problems, where the control input is bounded and we have at least 1 policy with bounded energy, this assumption is satisfied. If the system is linear, having a stabilizing controller is enough [4], so perhaps something like this could be enough also for our case.
>
>
>
>
>
> **References**
> 1. Kakade, Sham, et al. "Information theoretic regret bounds for online nonlinear control." Advances in Neural Information Processing Systems 33 (2020)
>
> 2. Sukhija, Bhavya, et al. "Optimistic active exploration of dynamical systems." Advances in Neural Information Processing Systems 36 (2023)
>
> 3. Chua, K., et al. (2018) Deep reinforcement learning in a handful of trials using probabilistic dynamics models. Advances in neural information processing systems.
>
> 4. Simchowitz, Max, and Dylan Foster. "Naive exploration is optimal for online lqr." International Conference on Machine Learning. PMLR, 2020.

---

> > ### Author Response · Authors · 2024-08-09
> > **Follow up on the rebuttal**
> >
> > Dear Reviewer,
> >
> > We hope this message finds you well. We have noticed that our detailed rebuttal, addressing each of the concerns raised in your review, has not yet received any feedback. Please let us know if our responses address your concerns or if you have any other concerns. We hope with our response we could further reinforce your confidence and positive evaluation of our work.
> >
> > Thank you,
> >
> > Authors

---

> > > ### Comment · Reviewer_WMig · 2024-08-11
> > >
> > > Thanks to the author for the detailed response. For the open topics mentioned, I hope the authors will add some discussions in the final version to attract researchers from different fields. I will keep my score.

---

> > > > ### Author Response · Authors · 2024-08-13
> > > > **Official Comment**
> > > >
> > > > Thanks a lot for your active engagement in the review process. We will adapt the paper based on the feedback and discussion from the rebuttal period.

---

### Official Review · Reviewer_17bq · 2024-07-11

**Soundness:** 3
**Presentation:** 3
**Contribution:** 3
**Rating:** 6
**Confidence:** 2

**Summary:**

This work introduces a novel model-based RL algorithm, named NeoRL, for nonepisodic RL problems with unknown system dynamics and known costs. A cumulative regret bound is provided for NeoRL with well-calibrated dynamic models. The proposed method achieved lower accumulative regret and average cost compared with baselines in several continuous control tasks. In the end, NeoRL shows it needs less reset before convergence in a reverted pendulum task with automatic reset.

**Strengths:**

1. the research question this paper tries to address is important and interesting, as the difficulty of resetting the environment is a notorious blocker for deploying RL agents in the real world.

2. A bound for cumulative regret is provided for the proposed algorithm when certain conditions are satisfied.

3. The proposed algorithms show lower average cost and cumulative regret than baselines in the empirical evaluation.

**Weaknesses:**

1. The paper is difficult to follow for readers without much control theory background (such as me), and it is difficult to distinguish the algorithmic contributions of this work. Could the author provide a more intuitive explanation about how NeoRL is connected and different from [1,2] and which part of the algorithm actually improves performance in a non-episodic setting?

2. Another major concern of mine is the selection of baseline algorithms, which are not compared to other recent model-based RL algorithms, such as [3-5] and RL algorithms designed to deal with non-episodic problems [6, 7].

3. No analyses were presented to understand how critical design choices influence the proposed methods' performance, such as the choice of planning horizon H_0 and whether to double the planning horizon.

[1] Treven, L., Hübotter, J., Dorfler, F. and Krause, A., 2024. Efficient
exploration in continuous-time model-based reinforcement learning. *Advances in Neural Information Processing Systems*, *36*.

[2]Curi S, Berkenkamp F, Krause A. Efficient model-based reinforcement learning through optimistic policy search and planning. Advances in Neural Information Processing Systems. 2020;33:14156-70.

[3]Hafner D, Pasukonis J, Ba J, Lillicrap T. Mastering diverse domains through world models. arXiv preprint arXiv:2301.04104. 2023 Jan 10.

[4] Hansen N, Wang X, Su H. Temporal difference learning for model predictive control. arXiv preprint arXiv:2203.04955. 2022 Mar 9.

[5] Hansen N, Su H, Wang X. Td-mpc2: Scalable, robust world models for continuous control. arXiv preprint arXiv:2310.16828. 2023 Oct 25.

[6] Sharma A, Ahmad R, Finn C. A state-distribution matching approach to non-episodic reinforcement learning. arXiv preprint arXiv:2205.05212. 2022 May 11.

[7] Chen A, Sharma A, Levine S, Finn C. You only live once: Single-life reinforcement learning. Advances in Neural Information Processing Systems. 2022 Dec 6;35:14784-97.

**Questions:**

1. Could the author provide a more intuitive explanation about which part of the algorithm improves performance in non-episodic settings?

2. Could the author explain why the planning horizon needs to be doubled after each “artificial episode”?

3. Is NeoRL extendable to problems with unknown rewards (cost functions)?

**Limitations:**

The limitations and assumptions for the theory results are discussed in section 2

---

> ### Author Rebuttal · Authors · 2024-08-05
>
> Thank you for your feedback. In the following, we respond to the weaknesses and questions raised by the reviewer.
>
> **W1**: Difficult to follow for people with limited knowledge of control theory and connection to other prior work on optimistic exploration.
>
> **A1**: While control theory plays a crucial role in our analysis, we understand that it can be tough to follow with limited prior knowledge. We’d be happy to address any specific questions on the manuscript regarding this or provide additional explanations in the parts of the papers if the reviewer believes they might help the reader. On the comparison to [1, 2], both algorithms leverage the concept of optimism but consider very different settings. [1] studies the episodic setting in discrete time, whereas [2] for continuous time. As the reviewer zoj1 also highlights, we are not the first to propose optimistic exploration for model-based RL, but, to the best of our knowledge, we are the first to study it in the context of nonlinear systems and nonepisodic RL/average cost criterion. Algorithmically, a key difference is that [1] optimizes the policy for a finite horizon, [2] for continuous-time and finite horizon, and both reset the environment after every episode. In both cases, the horizon $H$ is fixed. We optimize for the average cost criterion, where there is no notion of a horizon. Since the settings/problems are very different, we cannot quantify the difference in performance among the different methods. NeoRL in essence leverages the same idea of optimism as [1,2] but studies the much more challenging non-episodic setting.
>
> **W2**: Baselines
>
> **A2**: In our experimental evaluation, we focus only on model-based methods due to their sample efficiency. As we highlight in section 4, MBRL methods vary mostly in how the model is represented (GPs, BNNs, world models etc), the choice of policy optimizer, and how the dynamics are propagated to facilitate exploration. Methods [3-5] study the first two axes (representation, e.g., RSSMs or policy optimization TDMPC). Furthermore, they are developed for the episodic/discounted reward setting with POMDPs, we study (theoretically and practically) the average cost criterion with MDPs, therefore these methods significantly diverge from our setting.  Crucially, our contribution is for the third axis of differentiation, i.e., we show that the celebrated principle of optimism works for many cases in the nonepisodic setting. To this end, we study different dynamics propagation approaches such as mean sampling (this is also used in [3-5] where no epistemic uncertainty is considered), trajectory sampling (PETS), and Thompson sampling. To the best of our knowledge, these are the only approaches (along the third axis) typically considered in MBRL.
> Lastly, note that [6, 7] both assume access to prior data, which we do not. Furthermore, they are model-free methods whereas we focus on model-based approaches.
>
> **Q1**: Intuitive explanation about which part of the algorithm improves performance in non-episodic settings
>
> **A1**: We are unsure about what the reviewer means by “which part of the algorithm improves performance in non-episodic settings”. We would appreciate it if the reviewer could elaborate on the question further. However, we also provide a tentative response;
> The key contribution of our work is to show that optimism, which is often used in bandit optimization and episodic RL, also yields theoretical guarantees and good empirical performance for the non-episodic case. Hence, akin to the episodic setting [1, 2, 8] optimism is crucial for NeoRL's theoretical guarantees. We also refer the reviewer to our response to W1.
>
> **W3/Q2**: Doubling of the planning horizon.
>
> **A2**: Note that it is not the planning horizon, but the model update horizon that is doubled, i.e., the frequency of our model update is reduced as we run the algorithm for longer. The intuitive explanation for this is that the longer we run the algorithm, the more data we collect and our model gets more accurate. Thus it requires less regular updates. Also, by doubling the horizon, we also improve the quality of the data by reducing transient effects in the collected trajectories. In practice, we observe that having a fixed horizon also works very well. In this case, the choice of the horizon depends on the available compute, the shorter the horizon, the more often you update your model. Furthermore, note that also other algorithms for the non-episodic setting increase the “artificial horizon/episode length” [9, 10]. This is also common for bandit optimization (see [11]).
>
> **Q3**: Extension to unknown rewards (costs).
>
> **A3**: Yes, NeoRL can in principle be extended to this setting. This can be simply done by including the cost “as part of your dynamics”. Moreover, given $x_t, a_t$ our model predicts the augmented $x_{t+1}, c_{t}$ and the last elemented of the augmented is used in our cost function. Under similar continuity assumptions on the cost as for the dynamics, we can extend our analysis to this setting.
>
> Having addressed the reviewer’s questions, we would appreciate it if the reviewer would increase their score for our paper. For any remaining questions, we are happy to provide further clarification.
>
> **References**
>
> [1] -- [7] as listed by the reviewer.
>
> [8] Kakade, Sham, et al. "Information theoretic regret bounds for online nonlinear control." Advances in Neural Information Processing Systems 33 (2020)
>
> [9] Simchowitz, Max, and Dylan Foster. "Naive exploration is optimal for online lqr." International Conference on Machine Learning. PMLR, 2020.
>
> [10]  Auer, Peter, Thomas Jaksch, and Ronald Ortner. "Near-optimal regret bounds for reinforcement learning." Advances in neural information processing systems 21 (2008).
>
> [11] Besson, Lilian, and Emilie Kaufmann. "What doubling tricks can and can't do for multi-armed bandits." arXiv preprint arXiv:1803.06971 (2018).

---

> > ### Comment · Reviewer_17bq · 2024-08-08
> >
> > I thank the authors for their very detailed response, which addressed most of my concerns and made the contribution of this work clear. I raised my score from 4 to 6.

---

> > > ### Author Response · Authors · 2024-08-09
> > > **Response to reviewer**
> > >
> > > Thanks a lot for the active engagement in the review process and for increasing our score. We are glad we could adeptly address your concerns. If there are any other questions, that we can address to to further improve your score or confidence in our work, please let us know.

---

### Official Review · Reviewer_zoj1 · 2024-07-12

**Soundness:** 4
**Presentation:** 4
**Contribution:** 4
**Rating:** 8
**Confidence:** 3

**Summary:**

The paper proposes NeoRL for non-episodic RL with nonlinear dynamical systems. NeoRL has a first-of-its-kind regret bound for general nonlinear systems with Gaussian process dynamics. The paper also proposes a practical implementation of NeoRL with MPC, which significantly outperforms baseline algorithms.

**Strengths:**

The paper paper proposes NeoRL, which has a first-of-its-kind regret bound for general nonlinear systems with Gaussian process dynamics.

The paper also proposes a practical implementation of NeoRL with MPC, which significantly outperforms baseline algorithms.

The paper is very well-written and easy to follow.

While the basic idea of the algorithm is not novel, the paper considers a very important topic of average RL and has a large impact on the theory of average RL, I think.

**Weaknesses:**

I do not see any particular weakness in this paper. Maybe one weakness is that the tightness of the derived bound is unclear because there is no lower bound for the considered setting, as the authors also mentioned in Conclusion.

**Questions:**

- Is there any comment on the tightness of the regret?
- How well does NeoRL scale to high-dimensional environments such as Humanoid?

**Limitations:**

The paper discuss the limitation of the theoretical results. I do not see any potential negative societal impact.

---

> ### Author Rebuttal · Authors · 2024-08-05
>
> We thank the reviewer for their invaluable feedback. We are happy to hear that they also appreciate the significance of our work. Below, we have our responses to the questions.
>
> **Q1**: Is there any comment on the tightness of the regret?
>
> **A1**: We would have loved to give a lower bound, but as acknowledged by the reviewer, lower bounds do not exist for this and in fact even for the simpler episodic setting (c.f, [1, 2] for example). However, we are actively working towards bridging this gap. Particularly, there is some hope that the upper bound is tight. This is motivated from results on Gaussian process bandit optimization [3], where the tightness of the regret is shown by providing similar order lower bounds. Since our upper bounds are of similar order, and motivated through a similar analysis, as the ones derived in GP bandits, we have some hope that they are also tight (with respect to T). However, overall, this is still an open problem.
>
> **Q2**: How well does NeoRL scale to high-dimensional environments such as Humanoid?
>
> **A2**: NeoRL has the same limitations as any model-based RL algorithm such as planning in high-dimensional input spaces. Particularly, NeoRL has to, in addition to the control inputs, optimize over the hallucinated controls $\eta$. There are heuristics to replace a direct optimization over $\eta$ with a sampling-based approach (c.f., [1, 4]), which scales much better (e.g., [4] evaluate it on a 58D system). Lastly, we use the iCEM [5] optimizer for planning, which has demonstrated scalability on the humanoid task.
>
>
> **References**
> 1. Kakade, Sham, et al. "Information theoretic regret bounds for online nonlinear control." Advances in Neural Information Processing Systems 33 (2020)
>
> 2. Curi, Sebastian, Felix Berkenkamp, and Andreas Krause. "Efficient model-based reinforcement learning through optimistic policy search and planning." Advances in Neural Information Processing Systems 33 (2020)
>
> 3. Scarlett, Jonathan, Ilija Bogunovic, and Volkan Cevher. "Lower bounds on regret for noisy gaussian process bandit optimization." Conference on Learning Theory. PMLR, 2017.
>
> 4. Sukhija, Bhavya, et al. "Optimistic active exploration of dynamical systems." Advances in Neural Information Processing Systems 36 (2023)
>
> 5. Pinneri, Cristina, et al. "Sample-efficient cross-entropy method for real-time planning." Conference on Robot Learning. PMLR, 2021.

---

> > ### Comment · Reviewer_zoj1 · 2024-08-13
> >
> > Thank you very much for the rebuttal! I acknowledge that I read it.

---

> > > ### Author Response · Authors · 2024-08-13
> > > **Official Comment**
> > >
> > > Thank you also for your engagement in the review process and the constructive feedback!

---

### Decision · Program_Chairs · 2024-09-25

**Decision:**

Accept (spotlight)

**Comment:**

Solid result on non-episodic learning in the control setting with Gaussian dynamics. Reviewers like it, and I would mainly suggest that the authors try and improve the presentation for the broader RL (or at least RL theory) audience, given that they position it as an RL result. Also, while I appreciate the authors describing the differences in setting in great detail to position their contributions, a similar overview of the new technical aspects would be very useful, since optimisitic planning, truncation to an effective horizon for stable systems etc are relatively standard techniques, and it would be useful for a reader to get a proof overview with the main novel components.